# Quantitative Proteomic Approach Reveals Altered Metabolic Pathways in Response to the Inhibition of Lysine Deacetylases in A549 Cells under Normoxia and Hypoxia

**DOI:** 10.3390/ijms22073378

**Published:** 2021-03-25

**Authors:** Alfonso Martín-Bernabé, Josep Tarragó-Celada, Valérie Cunin, Sylvie Michelland, Roldán Cortés, Johann Poignant, Cyril Boyault, Walid Rachidi, Sandrine Bourgoin-Voillard, Marta Cascante, Michel Seve

**Affiliations:** 1LBFA et BEeSy, Université Grenoble Alpes, Inserm, U1055, CHU Grenoble Alpes, PROMETHEE Proteomic Platform, 38000 Grenoble, France; alfonso.martin.bernabe@ki.se (A.M.-B.); vcunin@chu-grenoble.fr (V.C.); smichelland@chu-grenoble.fr (S.M.); sandrine.bourgoin@univ-grenoble-alpes.fr (S.B.-V.); 2PROMETHEE Proteomic Platform, TIMC-IMAG, Université Grenoble Alpes, CNRS, Grenoble INP, CHU Grenoble Alpes, 38000 Grenoble, France; 3Department of Biochemistry and Molecular Biomedicine, Institute of Biomedicine of Universitat de Barcelona, Faculty of Biology, Universitat de Barcelona, 08028 Barcelona, Spain; jtarragocelada@ub.edu (J.T.-C.); roldancg@gmail.com (R.C.); 4Department of Oncology-Pathology, BioClinicum, Karolinska Institutet, 17164 Solna, Sweden; 5Centro de Investigación Biomédica en Red de Enfermedades Hepáticas y Digestivas (CIBEREHD), Instituto de Salud Carlos III (ISCIII), 28029 Madrid, Spain; 6Reckonect, Institute for Advanced Biosciences—IAB, Université Grenoble Alpes, Inserm, UMR_S 1209/CNRS UMR 5309, 38000 Grenoble, France; johann@reckonect.com (J.P.); cyril@reckonect.com (C.B.); 7SyMMES/CIBEST, Université Grenoble Alpes, UMR 5819 UGA-CNRS-CEA, 38000 Grenoble, France; walid.rachidi@univ-grenoble-alpes.fr; 8BIG-BGE, Université Grenoble Alpes, CEA, Inserm, U1038, 38000 Grenoble, France; 9Metabolomics Node at Spanish National Bioinformatics Institute (INB-ISCIII-ES-ELIXIR), Institute of Health Carlos III (ISCIII), 28029 Madrid, Spain

**Keywords:** cancer metabolism, lysine deacetylase inhibitors, hypoxia, NSCLC

## Abstract

Growing evidence is showing that acetylation plays an essential role in cancer, but studies on the impact of KDAC inhibition (KDACi) on the metabolic profile are still in their infancy. Here, we analyzed, by using an iTRAQ-based quantitative proteomics approach, the changes in the proteome of *KRAS*-mutated non-small cell lung cancer (NSCLC) A549 cells in response to trichostatin-A (TSA) and nicotinamide (NAM) under normoxia and hypoxia. Part of this response was further validated by molecular and biochemical analyses and correlated with the proliferation rates, apoptotic cell death, and activation of ROS scavenging mechanisms in opposition to the ROS production. Despite the differences among the KDAC inhibitors, up-regulation of glycolysis, TCA cycle, oxidative phosphorylation and fatty acid synthesis emerged as a common metabolic response underlying KDACi. We also observed that some of the KDACi effects at metabolic levels are enhanced under hypoxia. Furthermore, we used a drug repositioning machine learning approach to list candidate metabolic therapeutic agents for *KRAS* mutated NSCLC. Together, these results allow us to better understand the metabolic regulations underlying KDACi in NSCLC, taking into account the microenvironment of tumors related to hypoxia, and bring new insights for the future rational design of new therapies.

## 1. Introduction

Lung cancer is the leading cause of cancer-related death worldwide, with an estimated upward trend of 2.1 million new cases and 1.8 million deaths per year (approximately 18.4% of total cancer deaths) in 2018 [1]. Non-small cell lung cancer (NSCLC) represents about 80–85% of all lung cancer cases, with an overall 5-year survival rate of 19% [2]. Traditional therapies in NSCLC such as radiotherapy and platinum-based chemotherapy lack specificity and often cause severe side effects as they affect healthy cells [3]. To address this problem, targeted therapies and immunotherapies have emerged as a way to specifically target cancer cells. However, the therapeutic response may be limited as tumors are often heterogeneous, and some cell populations within the tumor can be resistant to the inhibition of the selected target [4]. Thus, targeted therapies and immunotherapies will not benefit patients harboring other molecular driver gene mutations, such as patients whose tumors harbor activating *KRAS* mutation that leads to constitutively active RAS signaling independent of upstream signals [5]. To date, clinical approaches targeting mutated *KRAS* have been unsuccessful [6]. Several studies have shown that mutations in *KRAS* play a critical role in metabolic reprogramming in multiple cancers, including lung cancer [7,8].

Generally, in cancer cells, metabolic reprogramming is considered to be one of the hallmarks of cancer disease allowing them to produce enough energy, reducing power and precursors required for growth and proliferation [9,10]. The general metabolic phenotype of cancer cells consists of elevated glycolysis and lactate production even under aerobic conditions; a phenomenon called the “Warburg effect” [11]. This switch in metabolism allows cancer cells to survive with a limited oxygen supply characteristic of the tumor microenvironment as they become less dependent on oxidative phosphorylation (OXPHOS) [12]. Furthermore, an enhanced glucose uptake favors the pentose phosphate pathway (PPP) flux to generate enough reducing power for antioxidant defense and intermediates for nucleotide synthesis [13,14]. Additionally, a higher glutamine uptake is also considered a major component of the general metabolic phenotype of tumor cells, providing an additional advantage in synthesizing amino acids, nucleotides, and lipids [15]. Although many cancers share similar metabolic adaptations, cancer cells rewire their metabolic programs in response to changes in the tumor microenvironment and oncogenic signals such as an activating *KRAS* mutation. Indeed, *KRAS* mutated NSCLC A549 cell line, which exhibits high resistance to current treatments, is characterized by specific metabolic adaptations that rely on glycolysis and PPP [16].

Multiple studies have demonstrated that the hypoxic tumor microenvironment plays a critical role in cancer progression and drug resistance [17,18,19,20]. Under hypoxia, cancer cells engage metabolic adaptation strategies to survive and growth by activating a relevant gene expression program through HIF-1α. The HIF-1α-dependent gene program involves the up-regulation of genes associated with increased glycolysis and lactate production, such as glucose transporters, glycolytic enzymes, and LDH-A [21]. Interestingly, HIF-1α is regulated by acetylation and deacetylation processes: The transcriptional activity of HIF-1α is repressed by KDAC activity, and also sirtuins have emerged as regulators of HIF-1α [17,19,20,21,22]

On the other hand, and given the increasing importance of the post-translational modifications in cancer and metabolism, the inhibition of lysine deacetylases (KDAC) has emerged in recent years as another promising therapeutic strategy in cancer [23,24,25]. Lysine deacetylase inhibitors (KDACIs) are used clinically to treat hematological malignancies [26] but have not demonstrated clinical benefit in solid tumors [27]. Current research focuses on developing new KDACIs and prospects for therapeutic application in cancer and other pathological conditions. Additionally, even though the use of KDACIs in NSCLC is less established, several studies using them either as monotherapy or in combination with other inhibitors has created a new therapeutic scenario offering the possibility to improve the effectiveness reducing resistance to current treatments [27,28,29,30,31].

KDACIs target different classes of KDACs, suggesting that they may have a different effect on gene expression, affecting key cellular processes that ultimately can lead to apoptotic cell death. The use of KDACIs such as trichostatin A (TSA), an inhibitor of classes I, II, and IV KDAC enzymes, and nicotinamide (NAM), an inhibitor of class III KDACs (also known as sirtuins), has been shown to exhibit significant antitumor activity in terms of cell proliferation, viability and apoptosis in cell models of lung cancer [32,33]. However, their impact on cancer metabolism has not been addressed in detail in these studies although acetylation is known to play a critical role in regulating metabolism [34]. Herein, we investigated the effect of inhibition of KDAC by using TSA and NAM on the global proteome of the active *KRAS*-mutant NSCLC A549 cell line, which exhibits high resistance to current therapies. The inhibition of KDAC allowed us to explore how protein acetylation status affects the metabolic profile, thus better elucidating the link between acetylation and metabolic reprogramming in cancer.

Several studies have demonstrated the utility of isobaric Tags for Relative and Absolute Quantitation (iTRAQ)-based quantitative proteomic approaches for global in-depth profiling of proteomes by measuring the relative protein abundance in cancer samples. The evaluation of the differences between proteomes from cancer samples cultured under different culture conditions might identify specific proteome signatures associated with tumor growth and survival [35]. One of such culture conditions is certainly hypoxia exposure, which has been shown to induce a substantial shift in the proteome supporting metabolic processes when oxygen is limiting [36,37,38,39,40,41].

In this study, we performed a quantitative iTRAQ-based proteomic experiment coupled with two-dimensional (2D) fractionation (OFFGEL/RP-nano-LC) and mass spectrometry (MS) analysis together with other metabolic measurements and enzyme activity assays. Furthermore, we used a drug repositioning strategy to identify potential therapeutic opportunities for existing drugs targeting the metabolic reprogramming induced by KDAC inhibition and hypoxia. Our results explored new mechanisms of metabolic adaptations that lead to a deeper understanding of the regulation of lung cancer metabolism under KDACIs and hypoxic conditions, which may contribute to the development and design of new cancer therapies.

## 2. Results

### 2.1. KDAC Inhibition Leads to Reduced Cell Proliferation by Inducing Apoptosis, Cell Cycle Arrest, and Oxidative Stress in A549 Cells

The overall proliferation of A549 cells varied among the different KDACI treatments (Figure 1A). Both TSA and TSA/NAM double-treated cells inhibited cell proliferation, and decreased cell viability, which correlates with the significantly increased cell death by apoptosis (Figure 1B). NAM-treated cells, instead, exhibited a lower growth rate than control cells, also accompanied by a significant increase in apoptosis. The TSA/NAM double treatment further induced apoptotic cell death by approximately 2-fold compared with TSA single treatment. Finally, the hypoxic treatment only limited cell proliferation, and KDACI-treated cells under hypoxia followed a similar cell proliferative and apoptosis pattern to treated cells under normoxia.

TSA-treated cells exhibited a high percentage of cells at G2/M phase, while NAM-treated cells showed a delayed progression through the G1 phase. The combination of TSA and NAM treatment led to a drastic cell cycle arrest at the G2/M phase and decreased S phase. A similar trend was observed in KDACI treatments under hypoxia (Appendix A). ROS generation increased substantially in A549 cells under KDAC inhibition (Appendix A) and was enhanced under hypoxia. These elevated ROS levels could be associated with the dysregulation of the antioxidant defense system, a possibility that is further explored below.

### 2.2. KDAC Inhibition Modulates the Tumor Phenotype through Changes in the Metabolic Profile

The TSA treatment did not significantly alter the glucose consumption and lactate production under normoxia (Figure 2A,B). In contrast, both the NAM and TSA/NAM double treatments showed a significantly decreased glucose uptake and lactate production compared with the control cells. Under hypoxia, the glucose consumption and lactate production rates were significantly different in the TSA, NAM, and TSA/NAM treatments with respect to control cells under normoxia (Figure 2A,B). The glutamine uptake was significantly increased only under the TSA treatment in both contexts of normoxia and hypoxia (Figure 2C).

The iTRAQ-based quantitative proteomic analysis was performed using a 4800 MALDI-TOF/TOF mass spectrometer (AB Sciex, Les Ulis, France) and allowed the quantification of 834 proteins from 2,710 peptides. This analysis evidenced dysregulation of several proteins related to metabolism upon the different KDACI treatments (Figure 3A and Appendix A). The Gene Ontology (GO) enrichment analysis of the dysregulated proteins allowed us to decipher the biological processes (Appendix A) and the protein information resource (PIR) keywords (Appendix A) related to these treatments. Overall, proteins related to the generation of energy and intracellular transport were up-regulated (Figure 3B), while the transcription and RNA processing were generally down-regulated. Furthermore, according to the PIR keywords enrichment analysis, approximately 60–70% of the proteins that were dysregulated in all the treatments were proteins modified by acetylation, phosphorylation, or both post-translational modifications (Appendix A).

Our iTRAQ analysis confirmed a metabolic reprogramming in A549 cells treated with KDACIs (Figure 3C). Upon TSA treatment, enzymes from glycolysis (fructose-bisphosphate aldolase C, ALDC and phosphofructokinase 1, PFK1), TCA cycle (2-oxoglutarate dehydrogenase, OGDH) were found to be significantly up-regulated, while lactate dehydrogenase B (LDH-B) was down-regulated. This pattern was confirmed by Western blot and enzyme activity analysis (Figure 4), although glucose uptake and lactate production were not altered. Furthermore, some enzymes involved in the synthesis of fatty acids were significantly up-regulated after TSA treatment (very-long-chain enoyl-CoA reductase, TECR, and long-chain-fatty-acid-CoA ligase, ACSL3), whereas the fatty aldehyde dehydrogenase (ALDH10) involved in fatty acid β*-*oxidation was down-regulated (Figure 3C). Interestingly, the spermidine synthase (SRM) levels, which catalyzes the synthesis of the polyamine spermidine, were strongly down-regulated (Figure 3C).

In the NAM treatment, the glycolytic enzymes PFK1 and alpha-enolase (ENO1) were significantly increased, together with TECR (Figure 3C and Figure 4A). A similar pattern to the TSA treatment was also found in the TSA/NAM combined treatment (Figure 3C). Although in this case, enzymes involved in glutamine metabolism such as glutaminase 1 (GLS1) and ornithine aminotransferase (OAT) were significantly up-regulated. Additionally, SRM was again strongly down-regulated in the double treatment at similar levels than in the TSA treatment.

### 2.3. The Metabolic Changes Observed in KDAC Inhibition Are Enhanced under Hypoxia

As expected, in all the treatments under hypoxia, the HIF-1α factor appeared to be overexpressed compared to normoxia (Figure 4C). The KDAC inhibition under hypoxia showed a similar behavior to the one observed in normoxia, although the metabolic changes resulted in a larger magnitude (Figure 5A). Similarly, as the analysis performed in normoxic conditions, the GO enrichment analysis on biological processes showed that proteins related to the generation of energy and intracellular transport were significantly enriched when cells were treated with TSA, NAM, and both compounds under hypoxia (Figure 5B). In contrast, proteins involved in transcription and translation processes were down-regulated. The PIR keywords enrichment analysis also showed that 60–70% of the dysregulated proteins were acetylated or phosphorylated (Appendix A).

Regarding the changes in the abundance of metabolic enzymes, the inhibition of TSA under hypoxia had similar effects as control cells under normoxia, although in this case, hypoxia elicited a different metabolic response affecting both glycolysis and mitochondrial respiration. Upon TSA treatment, glycolytic enzymes and mitochondrial enzymes such as succinate dehydrogenase complex subunit A (SDHA), which is part of the mitochondrial respiratory complex II, and cytochrome c oxidase subunits 6A1 and 2 (COX6A1 and COX2) of mitochondrial complex IV were significantly up-regulated (Figure 4C,D and Figure 5C). Fatty acid metabolism was also altered, and SRM was strongly down-regulated (Figure 5C). The sirtuin inhibition by NAM treatment under hypoxia showed few significant effects on metabolic enzymes. Similar to the TSA treatment, in the TSA/NAM double treatment under hypoxia, the levels of ALDC, pyruvate dehydrogenase subunit E1 (PDHE), COX6A1, and ACSL3 were significantly up-regulated, whereas LDH-B was significantly down-regulated (Figure 5C).

Furthermore, concerning the oxidative stress results reported above, the proteomic analysis showed the dysregulation of some enzymes related to the antioxidant defense system (Figure 3C and Figure 5C). The expression of thioredoxin domain-containing protein 17 (TXNDC17) was significantly up-regulated in all the KDACI treatments except for TSA under normoxia. Thioredoxin domain-containing protein 5 (TXNDC5) was also up-regulated by TSA/NAM treatment under normoxia. In addition, the expression of peroxiredoxin-1 (PRDX1) and -4 (PRDX4) resulted altered upon NAM treatment and hypoxia.

### 2.4. Chemicals Targeting Proteins Affected by KDAC Inhibition under Hypoxia

Machine learning based on the integration of large-scale omics data is an emerging approach for identifying new therapeutic targets, new molecules, and repurposing existing drugs [42,43,44]. We applied this approach to our proteomic data to reveal which chemicals are known to target proteins revealed, in our study, affected by KDAC inhibition under hypoxia in *KRAS* mutated NSCLC A549 cells. We classified 500 chemicals according to their link to lung cancer and our network enriched in the following cell metabolism processes: ATP metabolic process (GO:0046034), oxidation–reduction process (GO:0055114), carbohydrate metabolic process (GO:0005975), lipid metabolic process (GO:0006629), and cellular protein metabolic process (GO:0044267) (Appendix A). Chemicals with higher scores target our network more, while chemicals in lower ranks were more reported to have a connection with lung cancer (especially MESH: D002282 pulmonary adenocarcinoma) in the literature, clinical trials, or clinical care. In Figure 6, we selected the top 70 chemicals targeting the protein dysregulation network when KDACi treatment favored A549 cell apoptosis (i.e., for combined KDACi (TSA and NAM) treatment in hypoxia conditions) (Appendix A-bis). Each chemical’s score was higher than 100 confirming a substantial connection with our protein network and metabolic processes. Among the 70 chemicals that our machine learning approach proposed for targeting protein adaptation network of KDAC inhibitors (TSA and NAM) upon hypoxia, we found drugs used as anti-cancer agents for NSCLC and other cancers or known for some anti-cancer properties in NSCLC (such as metformin, gemcitabine, 5-Fluorouracil, paclitaxel, imatinib, doxorubincin, and tamoxifen) target similar protein network in KDAC inhibitors (TSA and NAM) under hypoxia and normoxia.

## 3. Discussion

In recent years, increasing evidence has demonstrated the involvement of acetylation in the metabolic reprogramming of cancer cells, which mediates tumorigenesis, tumor progression, and resistance to cancer therapies. The aberrant expression and activity of KDACs are postulated to be one of the drivers of this metabolic reprogramming [45].

KDACIs exhibit multiple antitumor activities, including tumor cell differentiation, growth arrest, autophagy, and apoptosis in various NSCLC cancer cell lines and tumor xenografts [32,33,46,47,48]. The activation of either the extrinsic or intrinsic apoptotic pathway is a key event involved in the antitumor activity of KDACIs, although many aspects of their mechanisms of action remain conflicting and unclear [49]. Consistent with previous studies, our results confirmed increased apoptosis and cell cycle arrest under the inhibition of KDAC activity in adenocarcinoma A549 cells [50,51,52,53]. Furthermore, a remarkable synergistic apoptotic response and cell cycle arrest were observed following the TSA and NAM combined treatment.

The inhibition of KDAC induces a general increase in the level of acetylated proteins [54]. The exposure of cells to KDACIs not only induces hyperacetylation of histones, which is typically associated with a general increase of transcriptional activity but also targets other proteins from different subcellular compartments that regulate the proteome through the transcriptome. Wu et al. reported that the increased acetylation level in A549 cells under suberoylanilide hydroxamic acid (SAHA) treatment is positively correlated with the down-regulation of the global proteome expression level due to the crosstalk between acetylation and ubiquitination [55]. Indeed, protein acetylation and ubiquitination sites have now been considered cancer driver mechanisms per se [56]. In our study, proteins involved in the proteasome activity such as the proteasome activator complex subunits 1 and 2 (PSME1 and PSME2) and the ubiquitin-like modifier-activating enzyme 1 (UBA1) were significantly up-regulated under both TSA and NAM treatments, suggesting a higher protein degradation. On the other hand, both transcription and translation processes were down-regulated in KDACI treatments according to the DAVID functional annotation analysis. Therefore, both TSA and NAM treatments might cause general protein degradation in A549 cells, which also correlates with the low-proliferative phenotype reported here. Previous studies indicated that KDACs, especially KDAC6, play an essential role in protein quality control mechanisms [57]. Thus, the inhibition of KDAC to maintain proteostasis has been proposed to have therapeutic potential in cancer [58]. Indeed, several studies have demonstrated the potential therapeutic value of combining proteostasis regulators such as KDACIs and proteasome inhibitors in cancer [59,60].

Mitochondrial function and ROS production, have been reported to be altered during *KRAS*-driven malignant transformation [61,62,63]. KDACIs have also been shown to generate ROS in cancer [64,65,66]. The excessive production of ROS results in cellular oxidative damage and ultimately cell death [67]. For that reason, tumor cells usually have an enhanced antioxidant capacity to combat ROS. In addition to the classical antioxidant enzymes, thiol-containing redox enzymes such as the family of thioredoxins (TRXs) are also expressed in human lungs [68]. In our proteomic approach, we identified significant alterations in several enzymes involved in the antioxidant defense system. Interestingly, the expression of TXNDC17 has been recently associated with paclitaxel resistance in ovarian and colorectal cancer [69,70]. The activation of the antioxidant defense requires the coordinated action of a number of sirtuins that work together with ROS scavenging and generating pathways to maintain ROS homeostasis. Altogether, the iTRAQ results indicate that activation of ROS scavenging mechanisms, most likely in response to increased ROS levels is involved in the KDAC inhibition response.

The inhibition of KDAC may increase the acetylation level of metabolic enzymes, thereby affecting their catalytic activity, substrate accessibility, or amount of enzyme [71]. Besides, fluctuations of acetyl-CoA levels due to the changes in protein acetylation can affect metabolic processes as acetyl-CoA is required for the TCA cycle and fatty acid biosynthesis [72]. The changes in the levels of metabolic enzymes found in the current study suggest a switch in the metabolic profile of A549 cells towards a higher capacity of mitochondrial OXPHOS. This metabolic change correlates with the proliferation rates and apoptotic cell death reported here and with the impaired antioxidant defense and increased ROS production, which was previously described in KDACIs treatment of *KRAS*-driven cancers [64,65,66].

As most cancer cell lines, NSCLC A549 cells tend to exhibit the Warburg effect by relying on both an enhanced glycolysis and production of lactate together with an enhanced mitochondrial metabolism relying on other sources such as glutamine [73,74]. In our analysis, inhibition of KDAC classes I, II, and IV showed a significant overexpression and increased activity of several glycolytic enzymes. However, this glycolytic response did not induce a proportional increase in lactate production, whereas LDH-B was down-regulated. Therefore, the conversion from pyruvate and lactate seems to be impaired by TSA, which implies that the inhibition of Zn^2+^-dependent KDAC classes could compromise not only the Warburg effect but also other cancer metabolic adaptations related to LDH-B such as mTOR hyperactivation or lysosome acidification and autophagy [75,76]. Furthermore, the higher levels of TCA cycle enzymes are consistent with the hypothesis that a low lactate production would favor oxidation of pyruvate from glycolysis to acetyl-CoA for entry into the TCA cycle under TSA treatment. Furthermore, the up-regulation of ACSL3, which has recently demonstrated to be essential for tumorigenesis in *KRAS*-driven lung cancer [77,78], may also be involved in activating mitochondrial respiration from fatty acids. On the other hand, sirtuin inhibition exhibited a lower effect on metabolic enzymes than the other classes of KDAC inhibition, where it seems that only the glycolytic pathway was affected. Although the up-regulation of PFK1 and ENO1, the observed net decrease of glucose uptake and lactate production supports the hypothesis that NAM treatment may inhibit glycolysis most likely by the inactivation of the AMPK/SIRT1 pathway, impairing the Warburg effect even more than TSA treatment. AMPK plays a critical role in stimulating glucose uptake, and the link between AMPK and SIRT1 has been described in various studies [22,79,80]. Some of the metabolic changes observed in TSA or NAM treatment alone were particularly acute in the double treatment, suggesting that the effect of targeting both Zn^2+^-dependent KDAC classes and sirtuins on metabolic enzymes was stronger than only inhibiting Zn^2+^-dependent KDAC classes.

The up-regulation of glycolytic enzymes in cancer cells under hypoxic conditions were consistent with previous comparative proteomic studies [37,38,39]. However, while a switch from OXPHOS to glycolysis for ATP production is considered a major cancer cell adaptation to hypoxia, it has been suggested as well that low oxygen concentration in hypoxic regions of tumors may not be limiting OXPHOS [81,82,83]. Thus, A549 cells under hypoxia may still retain the function of OXPHOS to generate ATP. Our study confirmed the overexpression of HIF-1α among all the treatments except TSA single treatments whose expression was down-regulated. It has been previously demonstrated that KDACIs such as TSA can degrade HIF-1α stimulated by hypoxia with variable efficiency in different tumor cell lines [84]. Interestingly, we found increased iTRAQ ratios of TCA cycle enzymes under KDAC classes I/II and IV inhibition in hypoxia compared with normoxia. Since the mitochondrial function is usually attenuated in response to HIF-1α and hypoxia, we assume that A549 cells may activate mitochondrial metabolism as an adaptive response to KDACIs. This is supported by the fact that TSA treatments repressed the induction of HIF-1α, probably allowing mitochondrial respiration. In contrast, we suggest that sirtuin inhibition may enhance HIF-1α response, which agrees with the activation of HIF-1α through acetylation by SIRT1 [85,86,87]. Therefore, the metabolic adaptation of cancer cells to hypoxia could be less affected by NAM treatment than by TSA treatment. Finally, the TSA and NAM double treatment under hypoxia might cause a controversial scenario where HIF-1α may be stimulated and repressed simultaneously at different levels. Interestingly, we found LDH-B significantly down-regulated, while pyruvate dehydrogenase (PDHE) resulted significantly up-regulated. Such evidence confirms our precedent interpretation where residual production and release of lactate is enough to maintain an increased glycolytic flux, meanwhile it allows the entry of pyruvate in the TCA cycle under KDAC inhibition and hypoxia, thus equally impairing the Warburg effect as in normoxia. In addition, COX2 and COX6A1 were highly probably increased to enhance OXPHOS and ROS level, even though the limited oxygen availability.

Our machine learning analysis revealed a list of chemotherapeutic agents, including doxorubicin, paclitaxel, etoposide, tamoxifen, bortezomib, 5-fluorouracil, methotrexate, imatinib, gemcitabine, and metformin that may target proteins affected by KDAC inhibition under hypoxia in *KRAS* mutated NSCLC A549 cells.

Doxorubicin induces cell death by regulating oxidative stress mediated through the formation of mitochondrial ROS [88,89,90]. In contrast, paclitaxel and etoposide trigger cell death by engaging the intrinsic mitochondrial pathway of apoptosis [91]. Tamoxifen is a selective estrogen receptor (ER) modulator used as a hormonal therapeutic agent to treat ER-positive breast cancer. Several ER-independent mechanisms that modulate metabolic pathways have been reported; for example, an AMPK activation induced by tamoxifen through inhibition of mitochondrial complex I leading to a glycolysis activation, alteration of fatty acid metabolism, and inhibition of the mTOR pathway and translation [92]. In ER-positive NSCLC, tamoxifen was found to play a negative role in the growth of ER-positive NSCLC alone [93] or used as an adjuvant EGFR-TKI treatment [94,95].

It should be noted that previous studies have demonstrated that hypoxia protects tumor cells from apoptosis induced by chemotherapeutic agents. For instance, it leads chemoresistance to doxorubicin and tamoxifen in NSCLC cells, including A549 cells, through the HIF pathway [19,96], thus limiting chances of successful treatment. However, hypoxia may have no effect on apoptosis triggered by etoposide in A549 cells, suggesting a cell type-specific effect to trigger apoptosis under hypoxia by different chemotherapeutic agents [97].

Bortezomib is a protease inhibitor currently approved to treat multiple myeloma and mantle cell lymphoma that has been studied in preclinical and clinical settings of lung cancer, showing potential benefit in combination therapies [98].

Our analysis also identified therapeutic agents that interfere with DNA synthesis, such as nucleoside analogs 5-fluorouracil and gemcitabine, and the nucleotide biosynthesis inhibitor methotrexate. Gemcitabine has been used to treat NSCLC, either in combination with cisplatin or carboplatin or as a single drug adjuvant treatment. By contrast, 5-fluorouracil and methotrexate have shown limited therapeutic benefit in NSCLC [99].

Imatinib is a TKI with clinical activity in the treatment of chronic myeloid leukemia and gastrointestinal stromal tumors [100]. In lung cancer models, several studies have shown that imatinib has indirect antitumor activity through the inhibitory effects on lung cancer-associated fibroblasts [101,102,103]. Interestingly, imatinib resistance can be mediated by a metabolic shift characterized by an up-regulation of OXPHOS but also in part by HIF1α-dependent up-regulation of glycolysis [104,105,106,107]. Therefore, the use of OXPHOS inhibitors may reverse imatinib resistance.

The standard antidiabetic agent metformin has already been investigated for NSCLC repositioning drugs in clinical trials for non-diabetic NSCLC treatment patients [ClinicalTrial.gov identifier: NCT02285855] as it decreases lung cancer incidence and mortality [108]. It also improved the progression-free survival of diabetic patients chemoradiotherapy for NSCLC [109]. Metformin has been reported to sensitize EGFR-TKI–resistant NSCLC through the inhibition of IL-6 or AMP-activated kinase signaling [110,111] and increase the radiosensitivity of NSCLC through ATM and AMPK [112]. In addition to its repressive effects on IGF-1R and PI3K/AKT/mTOR pathways, inhibition of tumor angiogenesis, aerobic glycolysis, DNA repair, activation of the antitumoral immunity, metformin showed benefic effects on cancer cells by inhibiting mitochondrial complex I and lipid/protein synthesis, regulating glycolysis, glucose level uptake and insulin/insulin-like growth factor signaling availability for tumor cells.

Additionally, synergistic antitumor interactions of KDAC inhibitors with the proposed drugs have been identified in different cancer types, including NSCLC, representing novel approaches potentially exploitable in therapy [113,114,115]. The anti-cancer properties in NSCLC treatments, their role in metabolic reprogramming, and the synergistic effects with KDAC inhibitors of these chemical agents support the purpose of considering the drugs proposed by our machine learning approach for targeting metabolic reprogramming of *KRAS*-mutated NSCLC treatment upon hypoxia conditions.

In our study, we comprehensively investigated the regulation of metabolic enzymes of the NSCLC A549 cell line under the effects of the KDAC inhibitors (TSA and NAM) in normoxic and hypoxic conditions by using an iTRAQ-based quantitative proteomic approach. A549 cells, which present mutation in *KRAS* and display the Warburg phenotype, are resistant to current lung cancer therapies. Besides the low proliferation behavior under KDAC inhibition, we found altered expression patterns in metabolic enzymes, which were exacerbated by hypoxia (Figure 7). This allowed profiling the metabolic heterogeneity of the NSCLC A549 cell line according to the oxygen level. Under KDAC classes I/II and IV inhibition, we observed that A549 cells stimulate oxidative metabolism and oxidative stress while glycolysis is increased, but lactate production is decreased. Sirtuin inhibition impaired both glucose uptake and lactate production, although the up-regulation of some glycolytic enzymes.

Therefore, we propose the inhibition of both KDAC classes I/II/IV and sirtuins as a valid strategy to explore cancer metabolic reprogramming, as their inhibition provides new insights into the metabolic adaptations of the A549 cell line that may help design new and more effective therapies. Moreover, our machine learning approach revealed which chemicals may target the metabolic adaptations observed by KDAC inhibition under hypoxia when apoptosis was favored. These chemicals should be further explored to evaluate the therapeutic efficacy in *KRAS* mutated NSCLC under hypoxia conditions Although this study brings new insights to elucidate the effects of hypoxic response upon KDACi treatments on metabolic reprogramming, it is important to mention that the present study did not assess whether wild-type *KRAS* cells undergo distinct metabolic reprogramming events upon KDAC inhibition. Thus, we certainly cannot claim that the metabolic reprogramming observed in the study is a common response among *KRAS* mutant NSCLC cells and differs from wild-type *KRAS* NSCLC cells. Further studies involving wild-type and *KRAS* mutant NSCLC cells might help understand the impact of *KRAS* mutation on the metabolic reprogramming upon KDAC inhibition.

## 4. Materials and Methods

### 4.1. Cell Culture

Human lung adenocarcinoma epithelial cell line A549 was obtained from American Type Culture Collection and cultured in Dulbecco’s modified Eagle’s medium (DMEM; Gibco) containing 10 mM glucose, 10% fetal bovine serum (Gibco 10270), 0.5% penicillin (50 U mL^−1^) and streptomycin (50 μg mL^−1^). Cells were seeded and incubated for 24 h at 37 °C in a humidified 5% CO_2_ incubator. After 24 h, A549 cells were treated with ^24h^IC_20_ concentrations of 1 µM TSA (Sigma-Aldrich), 20 mM NAM (Sigma-Aldrich) and its combination (1 µM TSA and 20 mM NAM). A549 cells were either maintained in normoxia or placed into hypoxia for 24 h of treatment. Cells incubated in medium without KDACIs served as control. Hypoxia was achieved by placing cells in a Whitley H35 Hypoxystation (Don Whitley Scientific, UK) hypoxic incubator flushed with 5% CO_2_ and 95% N_2_ until the O_2_ content reached 1%.

### 4.2. Cell Viability Assay

Viability tests of both KDACIs (TSA and NAM) were performed at 24 h to determine the respective ^24h^IC_50_ values (Appendix A). A549 cells were cultured on 96-well plates in the aforementioned culture conditions, adding TSA or NAM at different concentrations in six replicates for 24 h. The cell viability was determined by the MTT colorimetric assay [116] after 1 h of incubation with 0.5 mg mL^−1^ of MTT.

### 4.3. Cell Proliferation Assay

A549 cells were seeded in 6-well plates at the density of 6 × 10^3^ cells mL^−1^. Once cells were attached, they were treated with 1 µM TSA, 20 mM NAM separately and in combination under both normoxia and hypoxia for 24 h and 48 h. Untreated cells cultured under both normoxia and hypoxia were used as controls. After the incubations, cells were washed with PBS, trypsinized, centrifuged and resuspended in PBS. Cell proliferation was determined by direct cell counting with Scepter™ 2.0 Handheld Automated Cell Counter (Millipore, Burlington, MA, USA).

### 4.4. Apoptosis Assay

Apoptosis was tested by fluorescence-activated cell sorting (FACS) analysis using Annexin V and propidium iodide (PI) staining to differentiate non-apoptotic cells (Annexin V^−^, PI^−^) and late apoptotic/necrotic cells (Annexin V^+^, PI^+^) from early apoptotic cells (Annexin V^+^, PI^−^) [117]. A549 cells were seeded in 6-well plates at the density of 6 × 10^3^ cells mL^−1^ and treated as described above. After the incubation, cells were trypsinized, centrifuged and resuspended in binding buffer (10 mM HEPES/NaOH, pH 7.4, 140 mM NaCl and 2.5 mM CaCl_2_). Annexin V conjugated to fluorescein isothiocyanate (FITC) was added to each sample and incubated for 30 min in darkness. Following PI addition, cells were analyzed by a flow cytometer (Gallios, Beckman Coulter, Brea, CA, USA) using FlowJo software.

### 4.5. Cell Cycle Analysis

Cell cycle analysis was assayed by flow cytometry using FACS after treatments described above. Following 24 h of incubation, adherent cells were collected by centrifugation after trypsinization, washed with PBS, resuspended in 0.5 mL of PBS and fixed by dropwise addition of 4.5 mL ice-cold 70% (*v*/*v*) ethanol. Cells were fixed for at least 4 h at −20 °C. Then, cells were centrifuged, washed with ice-cold PBS and incubated in PBS containing 50 µg mL^−1^ propidium iodide (PI, Sigma–Aldrich), 20 µg mL^−1^ DNase-free RNase A (Roche) for 1 h at room temperature. FACS analysis was carried out in a flow cytometer (Gallios, Beckman Coulter, Brea, CA, USA) and data were analyzed using FlowJo software (see flow cytometry gating strategy in Appendix A).

### 4.6. Measurement of Extracellular Metabolites

The concentration of glucose, glutamine, and lactate in media was determined spectrophotometrically [118,119,120]. The media were collected at the beginning and end of incubations and measured using a Cobas Mira Plus chemistry analyzer (Horiba ABX, Montpellier, France) by monitoring the production of NADPH in the specific reactions at 340 nm. Glucose concentration was measured using the hexokinase (HK) and glucose-6-phosphate dehydrogenase (G6PDH) coupled enzymatic reactions (ABX PentraTM Glucose HK CP; A11A01667). Lactate concentration was determined based on the lactate dehydrogenase (LDH) reaction. Glutamine concentration was measured by means of the conversion of glutamate via the glutaminase (GLS) reaction and glutamate, in turn, was determined using the glutamate dehydrogenase reaction. The metabolite consumption/production rates were normalized according to the cell proliferation. Metabolite concentrations were expressed as µmol mL^−1^ × 10^6^ cells h^−1^.

### 4.7. Determination of Intracellular ROS levels

Total intracellular reactive oxygen species (ROS) levels were determined using flow cytometry after treatments with TSA (1 µM) and NAM (20 mM) alone or in combination using 2′-7′-dichlorodihydrofluorescein diacetate (H_2_DCFDA, Invitrogen) probe. A549 cells were incubated with 5 µM H_2_DCFA in PBS supplemented with 5.5 mM glucose for 30 min at 37 °C. Next, PBS was replaced with DMEM supplemented with 10% FBS and incubated for 45 min at 37 °C. After incubation, cells were trypsinized and resuspended in a solution consisted of 50 mM H_2_DCFDA with 20 µg ml^−1^ propidium iodide (PI, Sigma–Aldrich) for flow cytometry analysis (Cyan ADP analyzer, Dako Cytomation, Agilent Technologies, Santa Clara, CA, USA). Cells positive only for PI were considered as necrotic and excluded from the analysis. Data analysis was performed using FlowJo software.

### 4.8. Proteomic Analysis

The overall workflow of our quantitative proteomic approach for the study is illustrated in Figure 8. Briefly, our approach included a filter-aided sample preparation (FASP) step prior iTRAQ labeling of peptides, followed by a two-step fractionation of labeled peptides (OFFGEL IEF/RP-nano-LC), MALDI-MS/MS analysis, database search and quantitative iTRAQ analysis as already published [121,122].

### 4.9. Filter-Aided Sample Preparation for iTRAQ Quantitation

Following 24 h of treatment, A549 cells were washed twice with PBS, frozen in liquid nitrogen and stored at −80 °C. For protein extraction, cells were scraped off the plates and incubated for 10 min at 4 °C using lysis buffer (0.1 M Tris-HCl pH 7.5 and 4% sodium dodecyl sulfate) containing protease inhibitors (1X Halt Protease Inhibitor Cocktail, Thermo Scientific), phosphatase inhibitors (1×Phosphatase Inhibitor Cocktail, Sigma-Aldrich) and KDACIs (2.5 µM TSA and 20 mM NAM). Lysates were sonicated at 4 °C and centrifuged at 14,000× *g* for 10 min at room temperature. Protein concentration was determined using the bicinchoninic acid (BCA) method (Thermo Fisher Scientific) according to the manufacturer’s instructions. An equal amount of protein for each condition (1 mg) was digested using a filter-aided sample preparation (FASP) protocol [123] modified for iTRAQ-OFFGEL-LC-MS/MS analysis as proposed by Campone, et al. [124]. Briefly, protein samples were boiled for 5 min at 95 °C with 0.1 M dithiothreitol (DTT) in the lysis buffer described previously. Then, the buffer was changed to 8 M urea in 0.1 M Tris-HCl pH 8.5 using a Microcon^®^ 30 kDa filter (Millipore). Samples were incubated with 12 mM of methylmethanethiosulfate (MMTS) for 30 min at room temperature. Afterwards, the buffer was changed to 0.5 M triethylammonium bicarbonate (TEAB) pH 8. Digestion was performed in the same filter device using trypsin/lysine C mix (Promega) in a 1:40 (*w*/*w*) protease-to-protein ratio and incubated on a shaker for 16 h at 37 °C. Peptides eluted after digestion on the filter device were purified and desalted using Pierce^®^ C-18 Spin Columns (Thermo-Scientific) according to the manufacturer’s instructions. Peptides were washed with 1% acetonitrile (ACN), 0.1% trifluoroacetic acid (TFA) and eluted with 80% ACN and 0.1% TFA in HPLC grade water before vacuum-drying samples using a Speed-vac (Eppendorf, Hamburg, Germany). The amount of peptides was measured using the BCA method at appropriate dilutions.

#### 4.9.1. iTRAQ Labeling

The use of iTRAQ allows to differentially quantify the proteins from each sample in the mass spectrometry analysis [125]. Samples were labeled using the 8-plex iTRAQ Reagent Kit (Sciex, Les Ulis, France) following the manufacturer’s instructions. Equal amounts (100 µg) of peptides from each sample were labeled with each iTRAQ reagent as follows: The iTRAQ reporter ions of *m*/*z* 113.1 for control cells in normoxia, the *m*/*z* 114.1 for TSA treated cells in normoxia, the *m*/*z* 115.1 for NAM treated cells in normoxia, the *m*/*z* 116.1 for TSA/NAM double-treated cells in normoxia, the *m*/*z* 117.1 for control cells in hypoxia, the *m*/*z* 118.1 for TSA treated cells in hypoxia, the *m*/*z* 119.1 for NAM treated cells in hypoxia and the *m*/*z* 121.1 for double-treated cells in hypoxia. Then, all the samples were pooled, and the labeling reaction was stopped by evaporation in a vacuum concentrator before the 2D fractionation.

#### 4.9.2. Two-dimensional (2D) fractionation: Peptide OFFGEL Isoelectrofocusing and Reversed Phase Nano-liquid Chromatography

Peptide fractionation was performed in two dimensions: OFFGEL isoelectrofocusing for pI-based peptide separation followed with reverse phase nano-liquid chromatography (RP-nano-LC) for further separation based on peptide hydrophobicity. The peptide OFFGEL fractionation was performed on a 3100 OFFGEL Fractionator using the 24-well OFFGEL Kit linear pH 3–10 (Agilent Technologies, Les Ulis, France) according to the manufacturer’s instructions. The pooled samples (containing in total 800 µg of peptides) were resuspended in 3.6 mL of focusing OFFGEL buffer and loaded in each of the 24 wells. The IPG gel strip was rehydrated and 150 µL of sample was loaded into each well. Peptides were focused with a maximum voltage of 8000 V, 50 µA, and 200 W until 50 kVh was reached. Afterwards, the 24 fractions were collected into individual tubes, dried in a vacuum concentrator and stored at −20 °C. Previous to the RP-nano-LC fractionation, OFFGEL fractions were purified and desalted using C18 ZipTip^®^ columns (Merck Millipore, Molsheim, France), vacuum-dried and then resuspended in 2% (*v*/*v*) ACN and 0.05% (*v*/*v*) TFA loading buffer. The peptides were separated according to their hydrophobicity on Ultimate 3000 nano-HPLC system controlled by Chromeleon 7 software (Dionex/Thermo Scientific, The Netherlands). For each sample, peptides were trapped on a µ-Precolumn (300 µm i.d. × 5 mm, C18 PepMap100, 5 µm, 100 Å pore size; Thermo Scientific) in 2% (*v*/*v*) ACN and 0.05% (*v*/*v*) at a flow rate of 20 µL min^−1^ for 3 min. Afterwards, peptides were separated in the reversed phase nano-HPLC column (Acclaim PepMap300 75 µm, 15 cm, nanoViper C18, 2 µm, 100 Å pore size; Thermo Scientific) in a binary gradient of buffer A (0.05% (*v*/*v*) TFA) and buffer B (80% (*v*/*v*) ACN and 0.05% (*v*/*v*) TFA) at a flow rate of 0.3 µL min^−1^. The entire run last for 60 min and the nano-LC gradient was performed during 5–40 min (5–35 min, 12–46% of buffer B; 35–40 min, 46–62% of buffer B). Finally, the column was washed with buffer B at 40 min (62–94%), 40–50 min (94%) and re-equilibrated at 50 min (94–4%). The nano-HPLC eluted peptides of each OFFGEL fraction were spotted by a PROBOT MALDI spotting device controlled by the µCarrier 2.0 software (Dionex/Thermo Scientific/LC Packings, The Netherlands) onto a MALDI sample plate, with a spot collection time of 15 s resulting in 200 spots per fraction. Samples were spotted on MALDI plates in duplicates. The matrix (α-Cyano-4-hydroxycinnamic acid, HCCA, 2 mg mL^−1^ in 70% ACN and 0.1% TFA) was continuously added at a dosage speed of 0.9 µL min^−1^.

#### 4.9.3. MALDI-TOF/TOF Analysis

RP-nano-LC-MS/MS analysis of spotted peptides was performed by using a 4800 MALDI-TOF/TOF mass spectrometer (AB Sciex, Les Ulis, France) controlled by the 4000 Series Data Explorer software (V.3.5.3, AB Sciex, Les Ulis, France). The mass spectrometer was operated in positive ion reflector mode externally calibrated by using the Peptide Calibration Standard II (Bruker Daltonics, Champs sur Marne, France) with 50 ppm of mass tolerance. Each spectrum was recorded in the mass range of 700–4000 *m*/*z*. Up to 40 of the most intense ions per spot characterized by a signal/noise ratio ≥ 40 were considered for MS/MS analysis. Selected ions for MS/MS analysis were activated using CID (collision-induced dissociation) activation mode.

#### 4.9.4. Database Search and Quantitative iTRAQ Analysis

MS and MS/MS raw data were processed using ProteinPilot software (version 4.5, Sciex, Les Ulis, France) with the Paragon Algorithm (Sciex, Les Ulis, France). The analysis was performed using the human reference-proteome database UniProtKB/Swiss-Prot (release 2015_06; 20,206 protein entries; European Bioinformatics Institute, Hinxton, United Kingdom). After the identification and quantification of peptides, a statistical analysis with the R software package Isobar [126] was used to relatively quantify the levels of proteins present in the different conditions. Only proteins with high confidence identification (≥95% of peptide confidence level), positive “used score” of 1 and global False Discovery Rate (FDR) cutoff of 1%, were considered for the Isobar analysis. iTRAQ ratios and *p*-values were calculated by Isobar estimating both technical and biological variability using a Cauchy distribution. iTRAQ ratios of the different treatments were normalized to the control cells under normoxia (114:113; 115:113; 116:113; 117:113; 118:113; 119:113; 121:113). The level of proteins with iTRAQ ratios >1 and a *p*-value ≤ 0.05 was considered to be significantly increased whereas the level of proteins with iTRAQ ratios < 1 and a *p*-value ≤ 0.05 was considered to be significantly decreased.

### 4.10. Gene Ontology Enrichment Analysis

The Database for Annotation, Visualization, and Integrated Discovery was used to explore the functional annotation of significantly dysregulated proteins. The corresponding UniProtKB accession numbers of these proteins were imported into the Gene Functional Annotation Tool of DAVID bioinformatics resources v6.7 (https://david.ncifcrf.gov/, accessed on 23 March 2021) [127,128]. The human genome was set as background. The functional category Protein Information Resource (PIR) Keywords and the Biological Process (BP) term were taken into consideration.

### 4.11. Western Blot Analysis

Equal amounts of protein lysates (35 μg) were separated by 12% SDS polyacrylamide gel electrophoresis (SDS-PAGE) and transferred to a nitrocellulose membrane (0.45 μm pore size, Bio-Rad) in transfer buffer (25 mM Tris-HCl, 192 mM glycine and 20% (*v*/*v*) ethanol) using the Mini Trans-Blot Electrophoretic Transfer Cell (Bio-Rad) at 45 V for 2 h. Membranes were blocked with PBS-0.1% (*w/v*) Tween 20 containing 5% (*w/v*) nonfat dry milk for 1 h at room temperature. Primary and secondary antibodies were diluted in the same solution. After primary and secondary antibody incubations, the blots were washed three times with PBS-0.1% (*w/v*) Tween 20. Membranes were blotted with primary antibodies overnight at 4 °C. Primary antibodies used include rabbit anti-β-actin (PA1-183, Thermo Fisher Scientific, 1:500), rabbit anti-PGI (SAB 2100894, Sigma-Aldrich, 1:500), rabbit anti-PFK1 (AB170868, Abcam, 1:1000), rabbit anti-ALDC (PA5-27659, Thermo Fisher Scientific, 1:500), rabbit anti-GAPDH (SAB 2100894, Sigma-Aldrich, 1:500), rabbit anti-ENO1 (PA5-29660, Thermo Fisher Scientific, 1:500), rabbit anti-LDH-A (SAB 1100050, Sigma-Aldrich, 1:500) and rabbit anti-LDH-B (PA5-27505, Thermo Fisher Scientific, 1:500), rabbit anti-OGDH (PA5-28195, Thermo Fisher Scientific, 1:1000) and anti-HIF1α (clone 28B, sc-13515, Santa Cruz Biotechnology, 1:200). HRP labeled anti-mouse IgG (A8924, Sigma- Aldrich, 1:3000) and anti-rabbit IgG (7074, Cell signaling, 1:2000) were used as secondary antibodies. Horseradish peroxidase (HRP) activity was assessed with the Clarity Western ECL substrate (Bio-Rad Laboratories, Inc.) and visualized with ChemidocTM XRS+ system (Bio-Rad Laboratories, Inc.), except for HIF-1α, which was analyzed by film detection of chemiluminescence. The levels of proteins were normalized and relatively quantified according to β-actin levels in at least three independent experiments. Western blot ratios (WB ratios) of the different metabolic enzymes amongst treatments were normalized to untreated A549 control cells under normoxia. Image analysis and quantification by densitometric scanning of replicate blots were performed using Image LabTM software (version 4.1, Bio-Rad Laboratories, Hercules, CA, USA).

### 4.12. Enzyme Activities

A549 cells were rinsed with ice-cold PBS and scraped with lysis buffer 20 mM Tris-HCl pH 7.5 containing 1 mM DTT, 1 mM EDTA, 0.2% (*v*/*v*) Triton, 0.02% (*v*/*v*) sodium deoxycholate and supplemented with a protease inhibitor cocktail (Sigma-Aldrich). Cell lysates were disrupted by sonication using a titanium probe (Vibra-Cell^TM^, Sonics & Materials, Newtown, CO, USA) and immediately centrifuged at 12,000× *g* for 20 min at 4 °C. The supernatant was used for the determination of specific LDH and PFK-1 enzyme activities using a Cobas Mira Plus chemistry analyzer (Roche) in two independent experiments performed in triplicate. LDH activity in the forward reaction was measured in 100 mM KH_2_PO_4_/K_2_HPO_4_ (pH 7.4) containing 0.2 mM pyruvate and 0.2 mM NADH, and the mixture was incubated at 37 °C. PKF1 activity was measured in 62.5 mM Tris-HCl (pH 8.0) containing 94 mM KCl, 1.85 mM DTT, 0.24 mM ATP, 0.60 mM MgSO_4_, 0.24 mM fructose-6-phosphate, 43.2 U mL^−1^ aldolase, 15.6 U mL^−1^ triosephosphate isomerase and 2.4 U mL^−1^ glyceraldehyde 3-phosphate dehydrogenase. The mixture was incubated at 37 °C. The enzyme activities were normalized by protein content using BCA method (Thermo Fisher Scientific). Enzyme activities are expressed as milliunits per milligram (mU mg^−1^) of protein.

### 4.13. Network Analysis by Node Embeddings

Protein-protein interaction signaling networks were modelized throughout Reckonect process based on the machine learning algorithm node2vec [129]. This process produces for each protein of the network a vector in an embedding space. These vectors can be compared via a cosine distance to predict their proximity. To predict the proximity between two groups of proteins, we sum of the cosine distance (exceeding a threshold set at 0.5 in this article) in the cosine similarity matrix of the nodes embeddings of all proteins of the 2 groups. We model a disease (or a chemical) through proteins co-cited with the disease (or the chemical) in the scientific literature (bioassays and scientific articles). The more a protein is co-cited with the disease (or the chemical), more important its embedding vector will have in the representation of the disease (or the chemical). Finally, the proximity of a group of proteins and a disease (or chemical) can be predicted by comparing the embedding vectors of the protein group to the embedding vectors of the proteins co-cited in the literature with the disease (or the chemical) by considering the number of co-citations has a weight of the vectors.

Altogether, this machine learning method produces a proximity score that we used to rank therapeutic candidates. It should be distinguished from restricted enrichment methodologies. Noteworthy, it uses the recurrence frequency of therapeutic association with proteins, and learns a model in an unsupervised dependent manner. As a machine learning method, it relies on previously described statistical methods [129].

This process uses several databases of protein-protein interactions as well as databases of relationships between chemicals, diseases, biological processes and proteins. (ChEMBL [130], PubChem [131], PUBMED/MEDLINE, CTD [132], DGIdb [133], SIGNOR [134], UniProt [135], BioGRID [136], Complex Portal [137], IntAct [138], mentha [139], MINT [140], Reactome [141], and STRING [142]).

In our analysis, we took into account only proteins extracted by proteomic analysis that are involved in cell metabolism processes (ATP metabolic process (GO:0046034), oxidation–reduction process (GO:0055114), carbohydrate metabolic process (GO:0005975), lipid metabolic process (GO:0006629) and cellular protein metabolic process (GO:0044267)). We compared the different sets of proteins extracted by proteomic analysis with bronchoalveolar adenocarcinoma (MESH: D002282 pulmonary adenocarcinoma) (Appendix A). Drug repositioning was performed for each condition with phase 4 used drugs targeting each network of deregulated proteins upon TSA, NAM under hypoxic and normoxic treatments (Appendix A). Only the top 500 chemicals are presented here.

### 4.14. Experimental Design and Statistical Rationale

Three biological replicates were used in all experiments except for the proteomic analysis. The statistical analysis of the proteomic data described above in “Database search and quantitative iTRAQ analysis” was based on one experiment setup prior to confirm the dysregulation of eight proteins by three biological replicates through Western blot analysis. Statistical significance of differences between treatments was calculated by Student’s *t*-test.

## Figures and Tables

**Figure 1 ijms-22-03378-f001:**
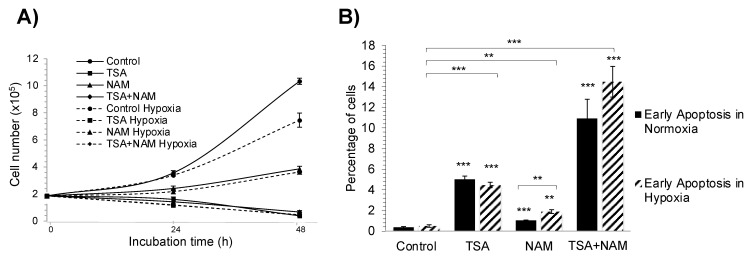
(**A**) Effect of trichostatin-A (TSA) and nicotinamide (NAM) treatments on A549 cell proliferation. A549 cells treated with 1 μM TSA, 20 mM NAM, or both 1 μM TSA and 20 mM NAM for 24 h and 48 h of incubation under normoxia or hypoxia. Dots represent means ± standard deviations of three independent experiments. (**B**) Apoptosis analysis of KDACI-treated A549 cells under normoxia and hypoxia. Apoptosis was measured after 24 h of incubation. Cells in the stage of early apoptosis are represented as the percentage with respect to total cells. A549 cells were treated with 1 μM TSA, 20 mM NAM or both 1 μM TSA and 20 mM NAM for 24 h under normoxia or hypoxia. Bars represent the means ± standard error of the mean of three independent experiments. The asterisks above bars indicate statistically significant differences compared to normoxic control cells. Asterisks above curly brackets indicate statistically significant differences between hypoxic and normoxic treatments and between hypoxic treatments and hypoxic control cells. Statistical significance was assessed by a two-tailed Student’s *t*-test. ** *p* ≤ 0.01; *** *p* ≤ 0.001.

**Figure 2 ijms-22-03378-f002:**
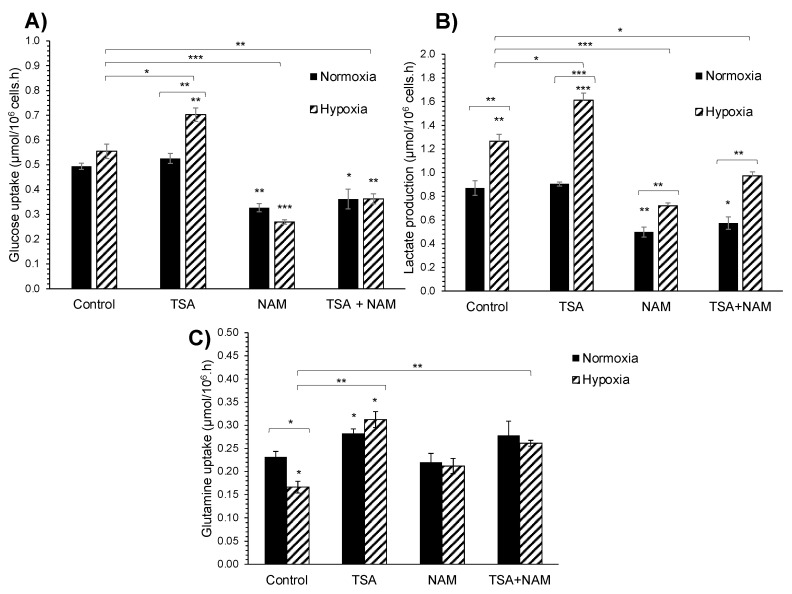
Extracellular metabolite quantitation of KDACI-treated A549 cells under normoxia and hypoxia. The glucose uptake (**A**), lactate production (**B**), and glutamine uptake (**C**) were measured in the beginning and at the end of the 24 h-incubation, and the metabolite consumption/production rates were normalized by the number of cells in each condition. (**A**–**C**) A549 cells were treated with 1 μM TSA, 20 mM NAM, or both 1 μM TSA and 20 mM NAM for 24 h under normoxia or hypoxia. Bars represent the means ± standard error of the mean of three independent experiments. The asterisks above bars indicate statistically significant differences compared to normoxic control cells. The asterisks above curly brackets indicate statistically significant differences between hypoxic and normoxic treatments and between hypoxic treatments and hypoxic control cells. Statistical significance was assessed by a two-tailed Student’s *t*-test. * *p* ≤ 0.05; ** *p* ≤ 0.01; *** *p* ≤ 0.001.

**Figure 3 ijms-22-03378-f003:**
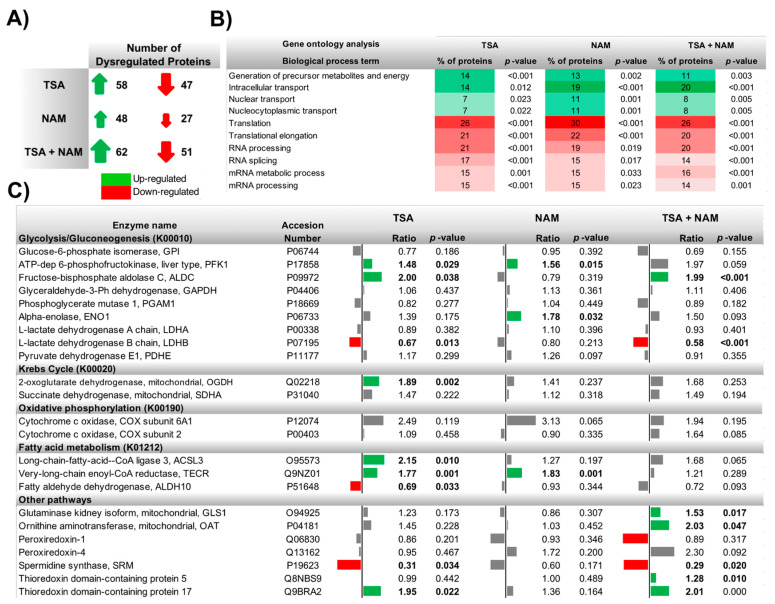
Effect of KDAC inhibition on the global proteome and metabolic enzymes compared to control A549 cells. Quantitative proteomic analysis of differentially expressed proteins in A549 cells treated with 1 μM of TSA, 20 mM of NAM or both 1 μM TSA, and 20 mM NAM for 24 h under normoxic conditions. (**A**) The number of up-regulated and down-regulated proteins (isobaric Tags for Relative and Absolute Quantitation (iTRAQ) ratio > 1 and <1, respectively) showing significant (*p*-value ≤ 0.05) differences between TSA, NAM and TSA/NAM treatments with respect to control cells. (**B**) GO enrichment analysis of the Biological process term for each condition shown as the percentage of proteins related to each process. All biological processes are shown as significantly (*p*-value ≤ 0.05) up-regulated or down-regulated. (**C**) Quantitative measurement of the main metabolic enzymes identified using the iTRAQ approach for the different conditions compared to untreated control cells. Significantly up-regulated enzymes (iTRAQ ratio > 1 and *p*-value ≤ 0.05) are represented in green and significantly down-regulated enzymes (iTRAQ ratio < 1 and *p*-value ≤ 0.05) are represented in red. Non-significantly up-regulated, and down-regulated enzymes are represented in gray.

**Figure 4 ijms-22-03378-f004:**
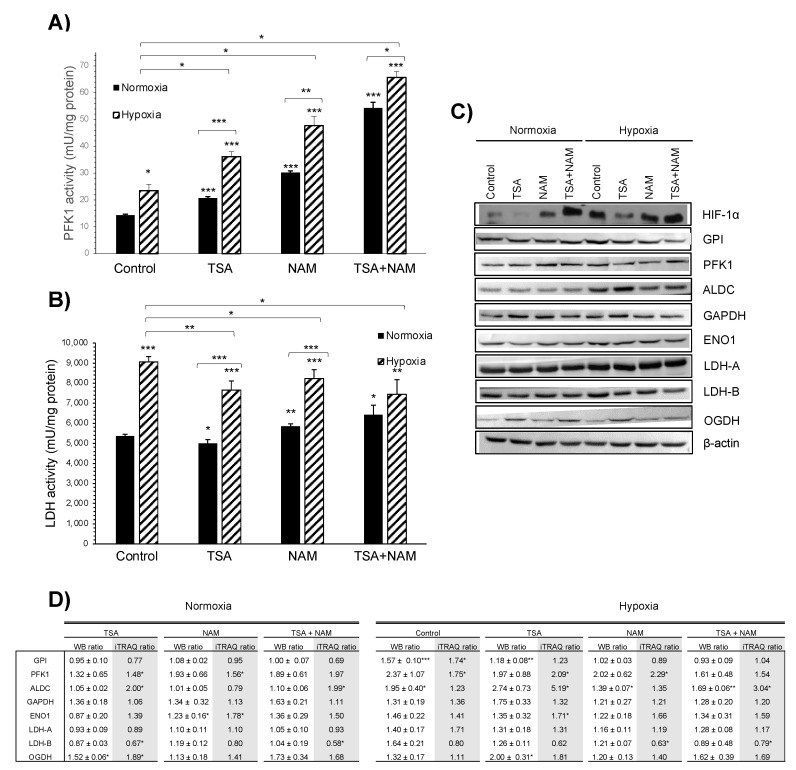
Effect of KDAC inhibition on enzyme activities in A549 cells under normoxia and hypoxia. (**A**,**B**) The ATP- dependent 6-phosphofructokinase (PFK1) (**A**) and lactate dehydrogenase (LDH) (**B**) enzymatic activities were measured after 24 h of incubation, and activities were normalized to intracellular protein content in each condition. A549 cells were treated with 1 μM of TSA, 20 mM of NAM, and both 1 μM TSA and 20 mM NAM for 24 h of incubation under normoxia and hypoxia. Cells incubated in medium without KDACIs served as control. Bars represent the means ± standard error of the mean of three independent experiments. The asterisks above bars indicate statistically significant differences compared to normoxic control cells. The asterisks above curly brackets indicate statistically significant differences between hypoxic and normoxic treatments and between hypoxic treatments and hypoxic control cells. Statistical significance was assessed by a two-tailed Student’s *t*-test. *, *p* ≤ 0.05; **, *p* ≤ 0.01; ***, *p* ≤ 0.001. (**C**) Western blot images of HIF-1α and selected metabolic enzymes identified by iTRAQ. A549 cells were treated with 1 μM of TSA, 20 mM of NAM and both 1 μM TSA and 20 mM NAM for 24 h of incubation under normoxia and hypoxia. Cells incubated in medium without KDACIs served as control. Immunoblotting of hypoxia-inducible factor-1α (HIF-1α), glucose-6-phosphate isomerase (GPI), phosphofructokinase-1 (PFK1), fructose-bisphosphate aldolase C (ALDC), glyceraldehyde-3-phosphate dehydrogenase (GAPDH), alpha-enolase (ENO1), lactate dehydrogenase A (LDH-A), lactate dehydrogenase B (LDH-B) and 2-oxoglutarate dehydrogenase (OGDH). β-actin was used as the loading control. (**D**) Densitometry analysis of selected metabolic enzymes shown in C. The ratios of the Western blot bands (WB ratios) of KDACI-treated cells to control cells under normoxia after normalization to β-actin. Densitometric values are presented as mean ± standard deviation. Asterisks indicate significant differences compared to untreated normoxic control cells assessed by two-tailed Student’s *t*-test in WB ratios and R software package Isobar (iTRAQ ratios) * *p* ≤ 0.05; ** *p* ≤ 0.01; *** *p* ≤ 0.001.

**Figure 5 ijms-22-03378-f005:**
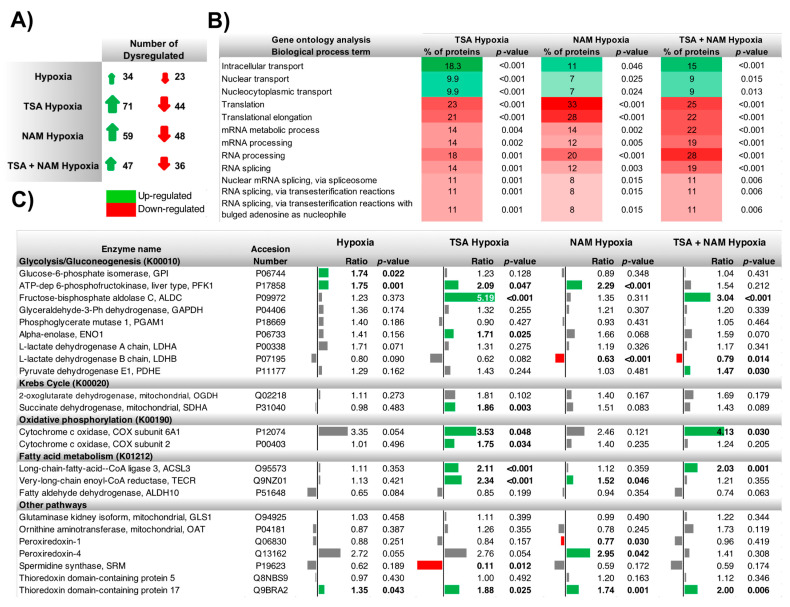
Effect of KDAC inhibition and hypoxia on the global proteome and metabolic enzymes compared to control A549 cells under normoxia. Quantitative proteomic analysis of differentially expressed proteins in A549 cells treated with 1 μM of TSA, 20 mM of NAM, and both 1 μM TSA and 20 mM NAM for 24 h under hypoxic conditions. (**A**) The number of up-regulated and down-regulated proteins (iTRAQ ratio > 1 and <1, respectively) showing significant (*p*-value ≤ 0.05) differences between TSA, NAM, and TSA/NAM treatments under hypoxia with respect to control cells under normoxia. (**B**) GO enrichment analysis of the Biological process term for each condition shown as the percentage of proteins related to each process. All biological processes are shown as significantly (*p*-value ≤ 0.05) up-regulated or down-regulated. (**C**) Quantitative measurement of the main metabolic enzymes identified using the iTRAQ approach for the different conditions compared to untreated control cells under normoxia. Significantly up-regulated enzymes (iTRAQ ratio > 1 and *p*-value ≤ 0.05) are represented in green and significantly down-regulated enzymes (iTRAQ ratio < 1 and *p*-value ≤ 0.05) are represented in red. Non-significantly up-regulated and down-regulated enzymes are represented in gray.

**Figure 6 ijms-22-03378-f006:**
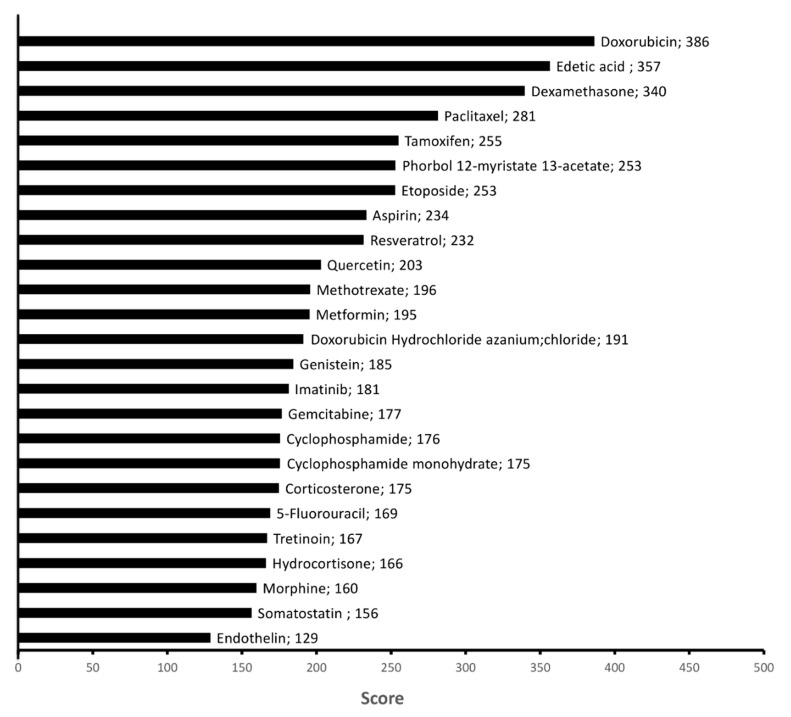
Classification of drugs targeting metabolic protein networks modulated by TSA and NAM in hypoxia conditions. The classification of chemicals is done through a deep machine learning according to their link to bronco-alveolar adenocarcinoma and cell metabolism processes (ATP metabolic process (GO:0046034), oxidation–reduction process (GO:0055114), carbohydrate metabolic process (GO:0005975), lipid metabolic process (GO:0006629), and cellular protein metabolic process (GO:0044267)). Five hundred chemicals were classified. A high score symbolized a high link of the chemical with the cell metabolism process and our protein network. The classification presented here was obtained by extracting drugs obtained by our machine learning analysis in the top 70 rank of chemicals related to MESH: D002282 pulmonary adenocarcinoma with exclusion of chemicals/metabolites. The top 70 rank ensures a high link between the chemical and data reported on these chemicals in the literature in broncho-alveolar adenocarcinoma (MESH: D002282 pulmonary adenocarcinoma).

**Figure 7 ijms-22-03378-f007:**
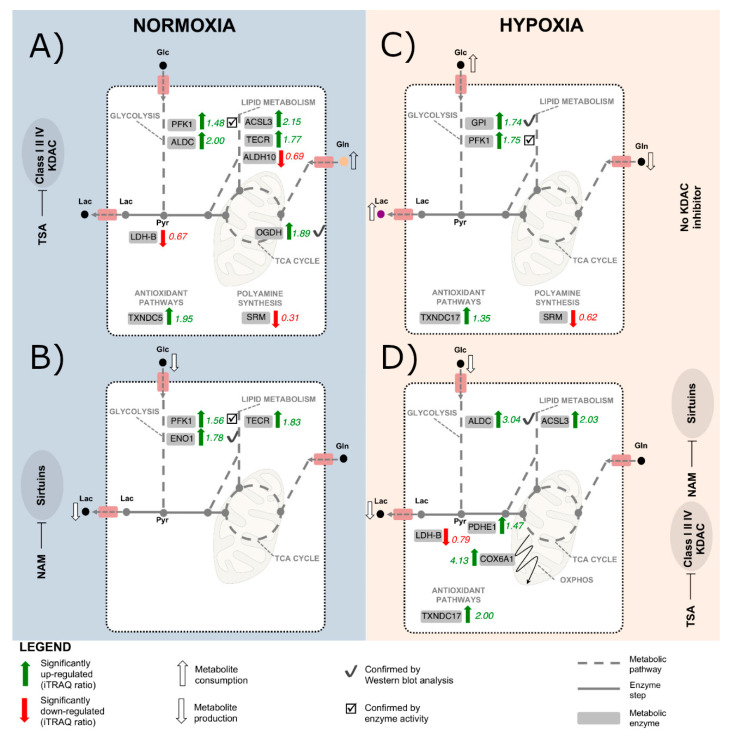
Schematic representation of the dysregulation of metabolic enzymes triggered by TSA and NAM under normoxia and hypoxia in A549 cells. Metabolic pathways are represented with dysregulated enzymes quantified by iTRAQ. Significantly up-regulated enzymes (iTRAQ ratio > 1 and *p*-value ≤ 0.05) are represented by green arrows. Significantly down-regulated enzymes (iTRAQ ratio < 1 and a *p*-value ≤ 0.05) are represented by red arrows. Non-significantly dysregulated enzymes are represented by grey arrows. Metabolic dysregulations confirmed by enzyme activities and Western blot analyses are represented by check box and check marks, respectively. Downward and upward white arrows indicate significant changes in metabolite consumption and production rates. (**A**) Metabolic enzyme profile in A549 cells under inhibition of classes I/II/IV KDAC by 1 μM of TSA for 24 h incubation under normoxic conditions. (**B**) Metabolic enzymes regulation in A549 cells under sirtuin inhibition by 20 mM NAM for 24 h incubation under normoxic conditions. (**C**) Metabolic enzymes regulation in A549 cells under hypoxic conditions for 24 h incubation. (**D**) Metabolic enzymes regulation in A549 cells under inhibition of both classes I/II/IV KDAC and sirtuins by 1 μM TSA and 20 mM NAM for 24 h incubation under hypoxic conditions. AcCoA: Acetyl-CoA; ACSL3: Long-chain-fatty-acid-CoA ligase 3; ALDC: Fructose-bisphosphate aldolase C; ALDH10: Fatty aldehyde dehydrogenase; COX6A1: Cytochrome c oxidase subunit 6A1; ENO1: Alpha-enolase; Glc: Glucose; Gln: Glutamine; KDAC: Lysine deacetylases; Lac: Lactate; LDH-B: Lactate dehydrogenase B; NAM: Nicotinamide; OGDH: 2-oxoglutarate dehydrogenase, mitochondrial; OXPHOS: Oxidative Phosphorylation; Pyr: Pyruvate; PDHE: Pyruvate dehydrogenase E1; PFK1: ATP-dependent 6-phosphofructokinase 1, liver type; Pyr: Pyruvate; SIRT: Sirtuins; SRM: Spermidine synthase; TCA: tricarboxylic acid; TECR: Very-long-chain enoyl-CoA reductase; TSA: Trichostatin A; TXNDC17: Thioredoxin domain-containing protein 17; TXNDC5: Thioredoxin domain-containing protein Our machine learning analysis revealed a list of chemotherapeutic agents, including doxorubicin, paclitaxel, etoposide, tamoxifen, bortezomib, 5-fluorouracil, methotrexate, imatinib, gemcitabine and metformin that may target proteins affected by KDAC inhibition under hypoxia in *KRAS* mutated NSCLC A549 cells.

**Figure 8 ijms-22-03378-f008:**
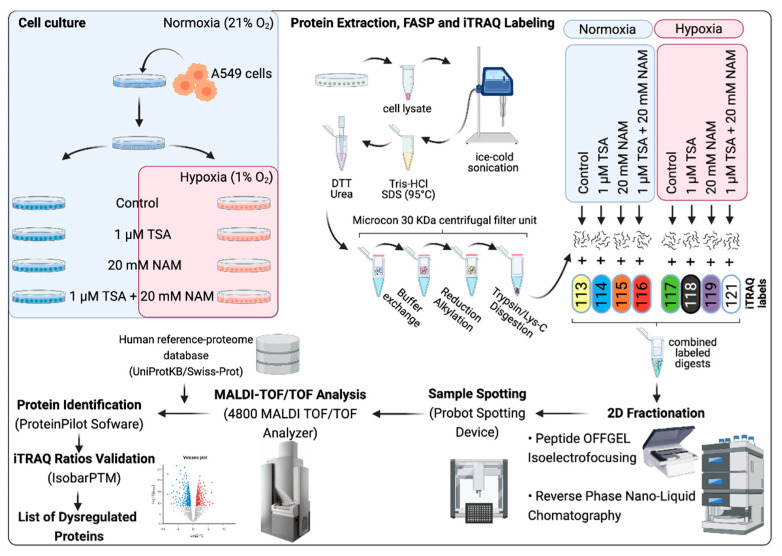
Schematic workflow of the proteomic approach used in the present study. A549 cells were treated with 1 μM of TSA, 20 mM of NAM and both 1 μM TSA and 20 mM NAM for 24 h under normoxia and hypoxia. Cells incubated in medium without KDACIs served as control. Cell lysates were processed according to the filter-aided sample preparation (FASP) protocol. Digested peptides of each treatment group were labeled with iTRAQ tags and separated in a 2-step fractionation. Fraction peptides were spotted on MALDI plates using a spotting system. Mass spectrometer 4800 MALDI TOF/TOF analyzer was used to collect MS and MS/MS data for identify and quantify proteins. iTRAQ ratios were quantified and validated. Proteins with iTRAQ ratios > 1 and a *p*-value ≤ 0.05 were considered to be significantly increased whereas proteins with iTRAQ ratios < 1 and a *p*-value ≤ 0.05 were considered to be significantly decreased.

## Data Availability

The proteomic data were deposited to the ProteomeXchange Consortium with the MassIVE identifier MSV000083174 (http://massive.ucsd.edu (accessed on 23 March 2021)) and ProteomeXchange identifier PXD011900 (http://www.proteomexchange.org (accessed on 23 March 2021)).

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
