# Peer review of "Quantitative Proteomic Approach Reveals Altered Metabolic Pathways in Response to the Inhibition of Lysine Deacetylases in A549 Cells under Normoxia and Hypoxia"

_ijms, 2021, doi:10.3390/ijms22073378_

Round 1

Reviewer 1 Report

Martin-Bernabe et al present a proteome of A549 cells treated with combinations of KDAC inhibitors (trichostatin-A (TSA) and nicotinamide (NAM)) under normoxic and hypoxic conditions. The authors observed clear phenotypes in growth rates, cell cycle arrest, and metabolite uptakes. The dysregulated proteins were validated with western blot analysis. Upon subsequent bioinformatics analysis, the authors identified drug target panels that may demonstrate therapeutic effects.

While the manuscript provides sufficient data to support the conclusion, the authors are advised to put less emphasis on the contribution of the KRAS mutation to the metabolic phenotypes, since there are no cell lines with normal KRAS as the control. The authors are also advised to better elaborate on the bioinformatics analysis in Figure 6. For instance, since the analysis is based on co-cited occurrence between the drug and the protein in literature, it would be advisable to show the literature/drug database they used in the Method section. Also, a brief description of how the authors' workaround with the multiple testing correction in the undisclosed algorithm will be appreciated.

In Figure 1B, the authors are recommended to provide a flow cytometry gating strategy in the supplementary information. Since the cells are undergoing apoptosis, it would be interesting to see if there is a sub-G1 phase cell cycle analysis. (Figure S2)

In Figure 2 (and all other figures), the error bar normally shows the standard error of the mean (SEM) instead of the standard deviation. It would be ideal to make the comparison according to the normoxia/hypoxia conditions, e.g. (normoxia TSA vs normoxia Control) and (hypoxia TSA vs hypoxia Control).

When presenting the proteomics data, the model of the mass spectrometer used, the number of proteins and peptides identified are displayed in the result section as a gauge of the data quality.

In Figure 4C, the western blots would be more convincing and readable if they can show the relative quantitation of the protein with the response to the control in the respective conditions.  

Author Response

Dear reviewers and editors,

Thank you very much for the revision of our manuscript. The authors sincerely appreciate the reviewers’ and editors’ efforts to make this study more complete, robust and valuable. We appreciate your feedback to strengthen the article and the time you took to review our research.

We addressed each of the reviewers points to improve the manuscript. Detailed responses to reviewers’ comments and major changes for easy inspection are listed below.

Following reviewers’ comments, we carefully revised the manuscript for grammar and style errors. The new changes have been identified in the word file with track changes.

We hope that the revised version of the manuscript meets your expectations and publication standards.

Sincerely yours,

Michel SEVE

Reviewer 1:

Comment #1: “Martin-Bernabe et al present a proteome of A549 cells treated with combinations of KDAC inhibitors (trichostatin-A (TSA) and nicotinamide (NAM)) under normoxic and hypoxic conditions. The authors observed clear phenotypes in growth rates, cell cycle arrest, and metabolite uptakes. The dysregulated proteins were validated with western blot analysis. Upon subsequent bioinformatics analysis, the authors identified drug target panels that may demonstrate therapeutic effects”.

Response: The authors thank the reviewer for the positive feedback.

Comment #2: “While the manuscript provides sufficient data to support the conclusion, the authors are advised to put less emphasis on the contribution of the KRAS mutation to the metabolic phenotypes, since there are no cell lines with normal KRAS as the control”.

Response: We thank the reviewer for point this out to us. We used the KRAS-mutated A549 cells as a well-studied cell line which proliferation is driven by the over-activation of the Ras-Raf-MEK-ERK pathway. However, we recognised that this study did not include any cell lines with wild-type KRAS, so the extend of KRAS mutation impact remains unclear. Therefore, we have added the following text to the discussion:

In Discussion – lines 751-758 (page 17):

We added: “Although this study brings new insights to elucidate the effects of hypoxic response upon KDACi treatments on metabolic reprogramming, it is important to mention that the present study did not assess whether wild-type KRAS cells undergo distinct metabolic reprogramming events upon KDAC inhibition. Thus, we certainly cannot claim that the metabolic reprogramming observed in the study is a common response among KRAS mutant NSCLC cells and differs from wild-type KRAS NSCLC cells. Further studies involving wild-type and KRAS mutant NSCLC cells might determine the impact of KRAS mutation on the metabolic reprogramming upon KDAC inhibition”.

Comment #3: The authors are also advised to better elaborate on the bioinformatics analysis in Figure 6. For instance, since the analysis is based on co-cited occurrence between the drug and the protein in literature, it would be advisable to show the literature/drug database they used in the Method section. Also, a brief description of how the authors' workaround with the multiple testing correction in the undisclosed algorithm will be appreciated.

Response: Thank you for giving us the opportunity to explain this bioinformatic analysis better. To respond to the reviewer concerns, we have provided further details of the analysis in the material and methods:

In Material and methods – lines 1028-1029 (page 23):

We replaced: “Protein-protein interaction signaling networks were modelized throughout Reckonect process [Poignant and Boyault, undisclosed]” by “Protein-protein interaction signaling network were modelized throughout the Reckonect process based on the machine learning algorithm node2vec [1]”.

In Material and methods – lines 1042-1056 (pages 23-24):

We added the following text:

“Altogether, this machine learning method produces a proximity score that we used to rank therapeutic candidates. It should be distinguished from restricted enrichment methodologies.  Noteworthy, it uses the recurrence frequency of therapeutic association with proteins, and learns a model in an unsupervised dependent manner. As a machine learning method, it relies on precedently described statistical methods [1]. 

This process uses several databases of protein-protein interactions as well as databases of relationships between chemicals, diseases, biological processes and proteins (ChEMBL [2], pubchem [3], PUBMED/MEDLINE, CTD [4], DGIdb [5], SIGNOR [6], UniProt [7], BioGRID [8], Complex Portal [9], IntAct[10], mentha [11], MINT [12], Reactome [13], STRING [14]).”

References

  1. node2vec: Scalable Feature Learning for Networks. A. Grover, J. Leskovec. ACM SIGKDD International Conference on Knowledge Discovery and Data Mining (KDD), 2016.
  2. Gaulton A, Hersey A, Nowotka M, Bento AP, Chambers J, Mendez D, Mutowo P, Atkinson F, Bellis LJ, Cibrián-Uhalte E, Davies M, Dedman N, Karlsson A, Magariños MP, Overington JP, Papadatos G, Smit I, Leach AR. (2017) 'The ChEMBL database in 2017.' Nucleic Acids Res., 45(D1) D945-D954.
  3. Fu G, Batchelor C, Dumontier M, Hastings J, Willighagen E, Bolton E. PubChemRDF: towards the semantic annotation of PubChem compound and substance databases. J Cheminform 2015 July 14; 7:34. eCollection 2015. [PubMed PMID: 26175801] doi: 10.1186/s13321-015-0084-4.
  4. Davis AP, Grondin CJ, Johnson RJ, Sciaky D, Wiegers J, Wiegers TC, Mattingly CJ The Comparative Toxicogenomics Database: update 2021. Nucleic Acids Res. 2020 Oct 17.
  5. Integration of the Drug–Gene Interaction Database (DGIdb 4.0) with open crowdsource efforts. Freshour S, Kiwala S, Cotto KC, Coffman AC, McMichael JF, Song J, Griffith M, Griffith OL, Wagner AH. Nucleic Acids Research. 2020 Nov 25; doi: https://doi.org/10.1093/nar/gkaa1084. PMID: 33237278
  6. Licata L, Lo Surdo P, Iannuccelli M, Palma A, Micarelli E, Perfetto L, Peluso D, Calderone A, Castagnoli L, Cesareni G. SIGNOR 2.0, the SIGnaling Network Open Resource 2.0: 2019 update. Nucleic Acids Res. 2020 Jan 8;48(D1):D504-D510. doi: 10.1093/nar/gkz949. PMID: 31665520; PMCID: PMC7145695.
  7. The UniProt Consortium, UniProt: the universal protein knowledgebase in 2021, Nucleic Acids Research, Volume 49, Issue D1, 8 January 2021, Pages D480–D489, https://doi.org/10.1093/nar/gkaa1100
  8. Oughtred R, Rust J, Chang C, Breitkreutz BJ, Stark C, Willems A, Boucher L, Leung G, Kolas N, Zhang F, Dolma S, Coulombe-Huntington J, Chatr-Aryamontri A, Dolinski K, Tyers M. The BioGRID database: A comprehensive biomedical resource of curated protein, genetic, and chemical interactions. Protein Sci. 2021 Jan;30(1):187-200. doi: 10.1002/pro.3978. Epub 2020 Nov 23. PMID: 33070389; PMCID: PMC7737760.
  9. Meldal BHM, Bye-A-Jee H, Gajdoš L, Hammerová Z, Horácková A, Melicher F, Perfetto L, Pokorný D, Lopez MR, Türková A, Wong ED, Xie Z, Casanova EB, Del-Toro N, Koch M, Porras P, Hermjakob H, Orchard S. Complex Portal 2018: extended content and enhanced visualization tools for macromolecular complexes. Nucleic Acids Res. 2019 Jan 8;47(D1):D550-D558. doi: 10.1093/nar/gky1001. PMID: 30357405; PMCID: PMC6323931.
  10. Orchard S, Ammari M, Aranda B, et al. The MIntAct project--IntAct as a common curation platform for 11 molecular interaction databases. Nucleic Acids Research. 2014 Jan;42(Database issue):D358-63. DOI: 10.1093/nar/gkt1115.
  11. mentha: a resource for browsing integrated protein-interaction networks Alberto Calderone, Luisa Castagnoli & Gianni Cesareni Nature Methods 10, 690 (2013). doi:10.1038/nmeth.2561
  12. Calderone A, Iannuccelli M, Peluso D, Licata L. Using the MINT Database to Search Protein Interactions. Curr Protoc Bioinformatics. 2020 Mar;69(1):e93. doi: 10.1002/cpbi.93. PMID: 31945268.
  13. Jassal B, Matthews L, Viteri G, Gong C, Lorente P, Fabregat A, Sidiropoulos K, Cook J, Gillespie M, Haw R, Loney F, May B, Milacic M, Rothfels K, Sevilla C, Shamovsky V, Shorser S, Varusai T, Weiser J, Wu G, Stein L, Hermjakob H, D'Eustachio P. The reactome pathway knowledgebase. Nucleic Acids Res. 2020 Jan 8;48(D1):D498-D503. doi: 10.1093/nar/gkz1031. PMID: 31691815; PMCID: PMC7145712.
  14. Szklarczyk D, Gable AL, Lyon D, Junge A, Wyder S, Huerta-Cepas J, Simonovic M, Doncheva NT, Morris JH, Bork P, Jensen LJ, Mering CV. STRING v11: protein-protein association networks with increased coverage, supporting functional discovery in genome-wide experimental datasets. Nucleic Acids Res. 2019 Jan 8;47(D1):D607-D613. doi: 10.1093/nar/gky1131. PMID: 30476243; PMCID: PMC6323986.

Comment #4: “In Figure 1B, the authors are recommended to provide a flow cytometry gating strategy in the supplementary information. Since the cells are undergoing apoptosis, it would be interesting to see if there is a sub-G1 phase cell cycle analysis. (Figure S2)”

Response: Following up on this recommendation, we have provided the gating strategy for the cell cycle analysis in the supplementary information (Figure S4). However, we were not able to detect a peak in the sub-G1 phase. According to our protocol, we analyzed the cell cycle distribution of living cells. Cells were washed with PBS, and probably a high proportion of late apoptotic cells and other components of sub-G1 peak were excluded from the cell cycle analysis. Please note that while we were replying the comment, we noticed an error in the cell cycle protocol.

In Results – lines 812-814 (page 18):

We replaced: “Following 24 h of incubation, both adherent and detached cells were collected by centrifugation after trypsinization, washed with PBS, resuspended in 0.5 mL of PBS and fixed by dropwise addition of 4.5 mL ice-cold 70% (v/v) ethanol”. by “Following 24 h of incubation, adherent cells were collected by centrifugation after trypsinization, washed with PBS, resuspended in 0.5 mL of PBS and fixed by dropwise addition of 4.5 mL ice-cold 70% (v/v) ethanol”.  

Comment #5: In Figure 2 (and all other figures), the error bar normally shows the standard error of the mean (SEM) instead of the standard deviation. It would be ideal to make the comparison according to the normoxia/hypoxia conditions, e.g. (normoxia TSA vs normoxia Control) and (hypoxia TSA vs hypoxia Control).

Response: We thank the reviewer for these comments. We have revised the manuscript and made modifications to figures to implement your suggestions. Changes in Figures (1, 2, 4, S2 and S3) and legends have been made accordingly. We replaced SD by SEM error bars. We indicated statistically significant differences between hypoxia treatments and hypoxia control. Statistically significant differences between treatments in normoxia and hypoxia were also included. 

Comment #6: When presenting the proteomics data, the model of the mass spectrometer used, the number of proteins and peptides identified are displayed in the result section as a gauge of the data quality.

Response: Thank you for this remark. We have added the mass spectrometer instrument used in our study and the number of quantified peptides in the text.

In Results – lines 234-238 (page 32):

We replaced: "The iTRAQ-based quantitative proteomic analysis allowed a deeper insight into the metabolic changes that took place during the different KDACI treatments. This was achieved by quantifying 834 proteins and evidencing the dysregulation of proteins related to metabolism (Figure 3A and Table S1)." by "The iTRAQ-based quantitative proteomic analysis was performed using a 4800 MALDI-TOF/TOF mass spectrometer (AB Sciex, Les Ulis, France) and allowed the quantification of 834 proteins from 2710 peptides. This analysis evidenced dysregulation of several proteins related to metabolism upon the different KDACI treatments (Figure 3A and Table S1)”.

Comment #7: In Figure 4C, the western blots would be more convincing and readable if they can show the relative quantitation of the protein with the response to the control in the respective conditions.  

Response: Thank you for the advice. We have modified the Figure 4 to improve its readability. We updated the Figure 4 by including the densitometric values (panel D) that were in supplemental data (previously Table S4). In supplemental data, Table S4 was deleted and Table S5 and S5-bis renamed to “Table S4 and Table S4-bis”.

*Editors

Comment #1:

Authors should discuss the results and how they can be interpreted from the perspective of previous studies and of the working hypotheses. The findings and their implications should be discussed in the broadest context possible. Future research directions may also be highlighted.

Response: We thank the editors for these instructions in the discussion section. We added the sentences below to further discuss our results in a broader context and propose more perspectives.

In Discussion – lines 592-596 (page 13):

We added: “The up-regulation of glycolytic enzymes in cancer cells under hypoxic conditions were consistent with previous comparative proteomic studies [36-38].”

We replaced: “it has been suggested as well that low oxygen concentration in hypoxic regions of tumors may not be limiting OXPHOS [80-82].” by ”However, while a switch from OXPHOS to glycolysis for ATP production is considered a major cancer cell adaptation to hypoxia, it has been suggested as well that low oxygen concentration in hypoxic regions of tumors may not be limiting OXPHOS [80-82].”

In Discussion – lines 662-665 (page 16):

We added: “Our machine learning analysis revealed a list of chemotherapeutic agents, including doxorubicin, paclitaxel, etoposide, tamoxifen, bortezomib, 5-fluorouracil, methotrexate, imatinib, gemcitabine and metformin that may target proteins affected by KDAC inhibition under hypoxia in KRAS mutated NSCLC A549 cells”.

In Discussion – lines 751-758 (page 17):

Although this study brings new insights to elucidate the effects of hypoxic response upon KDACi treatments on metabolic reprogramming, it is important to mention that the present study did not assess whether wild-type KRAS cells undergo distinct metabolic reprogramming events upon KDAC inhibition. Thus, we certainly cannot claim that the metabolic reprogramming observed in the study is a common response among KRAS mutant NSCLC cells and differs from wild-type KRAS NSCLC cells. Further studies involving wild-type and KRAS mutant NSCLC cells might help understand the impact of KRAS mutation on the metabolic reprogramming upon KDAC inhibition.

References

  1. Bush, J.T., et al., Quantitative MS-Based Proteomics: Comparing the MCF-7 Cellular Response to Hypoxia and a 2-Oxoglutarate Analogue. ChemBioChem, 2020. 21(11): p. 1647-1655.

  1. Song, Z., et al., Delineation of hypoxia-induced proteome shifts in osteosarcoma cells with different metastatic propensities. Scientific Reports, 2020. 10(1): p. 727.

  1. Zhang, K., et al., Proteome Analysis of Hypoxic Glioblastoma Cells Reveals Sequential Metabolic Adaptation of One-Carbon Metabolic Pathways. Molecular & cellular proteomics : MCP, 2017. 16(11): p. 1906-1921.

  1. Moreno-Sánchez, R., et al., The bioenergetics of cancer: is glycolysis the main ATP supplier in all tumor cells? BioFactors (Oxford, England), 2009. 35(2): p. 209-225.

  1. Li, P., et al., Redox homeostasis protects mitochondria through accelerating ROS conversion to enhance hypoxia resistance in cancer cells. Scientific Reports, 2016. 6(1): p. 22831.

  1. Zheng, J.I.E., Energy metabolism of cancer: Glycolysis versus oxidative phosphorylation (Review). Oncology Letters, 2012. 4(6): p. 1151-1157.

Reviewer 2 Report

The paper reports an interesting study to assign analysed and collected protein panel to the molecular metabolic pathways referred to the normoxia and hypoxia processes in model cell line – A549 cells of lung cancer.

The proteins analysed were involved in inhibition of lysine deacetylases in above mentioned normoxia and hypoxia.

The paper has fairly written declaration of the work´s aims, experimental, results and discussion.

The topic is actual – the proteomic lung cancer study is hot topic and does deserve an attention to be further examined and here considered to be published when achieving given remarks.

Please, examine the paper from the grammatical point of view.

The graphics/figures are well structured and fairly described, especially metabolic molecular pathways of normo- and hypoxia

REMARKS

INTRODUCTION (note)

Please, add brief info of general principle of proteomics and how this approach helps to release issues with normoxia/hypoxia in cancer cells

INTRODUCTION (note, 1st paragraph) - this I also consider important

Generally, in cells, metabolic reprogramming is considered to be one of the hallmarks of cancer disease allowing cancer cells to produce enough energy, reducing power and precursors required for growth and proliferation [9, https://doi.org/10.1515/biol-2019-0070]

MATERIALS AND METHODS (note)

Please, draw and display a proteomic workflow step-by-step, and add it as a one of the figure.

Author Response

Dear reviewers and editors,

Thank you very much for the revision of our manuscript. The authors sincerely appreciate the reviewers’ and editors’ efforts to make this study more complete, robust and valuable. We appreciate your feedback to strengthen the article and the time you took to review our research.

We addressed each of the reviewers points to improve the manuscript. Detailed responses to reviewers’ comments and major changes for easy inspection are listed below.

Following reviewers’ comments, we carefully revised the manuscript for grammar and style errors. The new changes have been identified in the word file with track changes.

We hope that the revised version of the manuscript meets your expectations and publication standards.

Sincerely yours,

Michel SEVE

Reviewer 2:

Comment #1:

“The paper reports an interesting study to assign analysed and collected protein panel to the molecular metabolic pathways referred to the normoxia and hypoxia processes in model cell line – A549 cells of lung cancer.

The proteins analysed were involved in inhibition of lysine deacetylases in above mentioned normoxia and hypoxia.

The paper has fairly written declaration of the work´s aims, experimental, results and discussion.

The topic is actual – the proteomic lung cancer study is hot topic and does deserve an attention to be further examined and here considered to be published when achieving given remarks.

Please, examine the paper from the grammatical point of view.

The graphics/figures are well structured and fairly described, especially metabolic molecular pathways of normo- and hypoxia”

Response: The authors thank the reviewer for the positive feedback. As reported above, the manuscript has been thoroughly checked and edited to ensure that there is no grammatical or spelling mistakes.

Comment #2:

“INTRODUCTION (note)

Please, add brief info of general principle of proteomics and how this approach helps to release issues with normoxia/hypoxia in cancer cells

INTRODUCTION (note, 1st paragraph) - this I also consider important

Generally, in cells, metabolic reprogramming is considered to be one of the hallmarks of cancer disease allowing cancer cells to produce enough energy, reducing power and precursors required for growth and proliferation [9, https://doi.org/10.1515/biol-2019-0070]”

Response: The authors thank the reviewer for these comments. We agree that further information covering these points would be interesting. Following reviewer’s suggestion, we have improved the introduction section and added the following text:

In Introduction – lines 135-142 (page 3):

“Several studies have demonstrated the utility of ‘isobaric Tags for Relative and Absolute Quantitation’ (iTRAQ)-based quantitative proteomic approaches for global in-depth profiling of proteomes by measuring the relative protein abundance in cancer samples. The evaluation of the differences between proteomes from cancer samples cultured under different culture conditions might identify specific proteome signatures associated with tumor growth and survival [34]. One of such culture conditions is certainly hypoxia exposure, which has been shown to induce a substantial shift in the proteome supporting metabolic processes when oxygen is limiting [35-40].”

References

  1. Cheung, C.H.Y. and H.-F. Juan, Quantitative proteomics in lung cancer. Journal of Biomedical Science, 2017. 24(1): p. 37.

  1. Vinaiphat, A., et al., Application of Advanced Mass Spectrometry-Based Proteomics to Study Hypoxia Driven Cancer Progression. Frontiers in Oncology, 2021. 11: p. 98.

  1. Bush, J.T., et al., Quantitative MS-Based Proteomics: Comparing the MCF-7 Cellular Response to Hypoxia and a 2-Oxoglutarate Analogue. ChemBioChem, 2020. 21(11): p. 1647-1655.

  1. Song, Z., et al., Delineation of hypoxia-induced proteome shifts in osteosarcoma cells with different metastatic propensities. Scientific Reports, 2020. 10(1): p. 727.

  1. Zhang, K., et al., Proteome Analysis of Hypoxic Glioblastoma Cells Reveals Sequential Metabolic Adaptation of One-Carbon Metabolic Pathways. Molecular & cellular proteomics : MCP, 2017. 16(11): p. 1906-1921.

  1. Djidja, M.-C., et al., Identification of Hypoxia-Regulated Proteins Using MALDI-Mass Spectrometry Imaging Combined with Quantitative Proteomics. Journal of Proteome Research, 2014. 13(5): p. 2297-2313.

  1. Bousquet, P.A., et al., Hypoxia Strongly Affects Mitochondrial Ribosomal Proteins and Translocases, as Shown by Quantitative Proteomics of HeLa Cells. International Journal of Proteomics, 2015. 2015: p. 678527.

Comment #3:

MATERIALS AND METHODS (note)

Please, draw and display a proteomic workflow step-by-step, and add it as a one of the figure.

Response: We appreciate your advice and we have added a diagram illustrating the quantitative proteomic workflow used in this study (Figure 8) (page 20). The figure shows an overview of the different steps of the proteomic analysis. Briefly, these steps included sample preparation, filter-aided sample preparation (FASP) step, iTRAQ labeling, sample fractionation, MALDI-TOF/TOF analysis, protein identification, validation of iTRAQ ratios and generation of lists of dys-regulated proteins.  We also included a schematic representation of the culture conditions for the different KDACI treatments under normoxia and hypoxia.

*Editors

Comment #1:

Authors should discuss the results and how they can be interpreted from the perspective of previous studies and of the working hypotheses. The findings and their implications should be discussed in the broadest context possible. Future research directions may also be highlighted.

Response: We thank the editors for these instructions in the discussion section. We added the sentences below to further discuss our results in a broader context and propose more perspectives.

In Discussion – lines 592-596 (page 13):

We added: “The up-regulation of glycolytic enzymes in cancer cells under hypoxic conditions were consistent with previous comparative proteomic studies [36-38].”

We replaced: “it has been suggested as well that low oxygen concentration in hypoxic regions of tumors may not be limiting OXPHOS [80-82].” by ”However, while a switch from OXPHOS to glycolysis for ATP production is considered a major cancer cell adaptation to hypoxia, it has been suggested as well that low oxygen concentration in hypoxic regions of tumors may not be limiting OXPHOS [80-82].”

In Discussion – lines 662-665 (page 16):

We added: “Our machine learning analysis revealed a list of chemotherapeutic agents, including doxorubicin, paclitaxel, etoposide, tamoxifen, bortezomib, 5-fluorouracil, methotrexate, imatinib, gemcitabine and metformin that may target proteins affected by KDAC inhibition under hypoxia in KRAS mutated NSCLC A549 cells”.

In Discussion – lines 751-758 (page 17):

Although this study brings new insights to elucidate the effects of hypoxic response upon KDACi treatments on metabolic reprogramming, it is important to mention that the present study did not assess whether wild-type KRAS cells undergo distinct metabolic reprogramming events upon KDAC inhibition. Thus, we certainly cannot claim that the metabolic reprogramming observed in the study is a common response among KRAS mutant NSCLC cells and differs from wild-type KRAS NSCLC cells. Further studies involving wild-type and KRAS mutant NSCLC cells might help understand the impact of KRAS mutation on the metabolic reprogramming upon KDAC inhibition.

References

  1. Bush, J.T., et al., Quantitative MS-Based Proteomics: Comparing the MCF-7 Cellular Response to Hypoxia and a 2-Oxoglutarate Analogue. ChemBioChem, 2020. 21(11): p. 1647-1655.

  1. Song, Z., et al., Delineation of hypoxia-induced proteome shifts in osteosarcoma cells with different metastatic propensities. Scientific Reports, 2020. 10(1): p. 727.

  1. Zhang, K., et al., Proteome Analysis of Hypoxic Glioblastoma Cells Reveals Sequential Metabolic Adaptation of One-Carbon Metabolic Pathways. Molecular & cellular proteomics : MCP, 2017. 16(11): p. 1906-1921.

  1. Moreno-Sánchez, R., et al., The bioenergetics of cancer: is glycolysis the main ATP supplier in all tumor cells? BioFactors (Oxford, England), 2009. 35(2): p. 209-225.

  1. Li, P., et al., Redox homeostasis protects mitochondria through accelerating ROS conversion to enhance hypoxia resistance in cancer cells. Scientific Reports, 2016. 6(1): p. 22831.

  1. Zheng, J.I.E., Energy metabolism of cancer: Glycolysis versus oxidative phosphorylation (Review). Oncology Letters, 2012. 4(6): p. 1151-1157.

Round 2

Reviewer 2 Report

Authors have accomplished almost all given remarks.

However, i  stress to cite suggested paper as it also contributes to the recent developments in the cancer-based proteomics followed by metabolic pathways analysis.

INTRODUCTION (note, 2st paragraph)

Generally, in cancer cells, metabolic reprogramming is considered to be one of the hallmarks of cancer disease allowing them to produce enough energy, reducing power and precursors required for growth and proliferation [9, https://doi.org/10.1515/biol-2019-0070]”

Author Response

Dear reviewer,

We thank the reviewer for the remark. We have introduced the reference https://doi.org/10.1515/biol-2019-0070 in the second paragraph of the introduction as suggested by the referee. Please note that this reference appears in the manuscript as reference number 10. We hope this version we are submitting now accomplish all the given remarks.

We thank the referee for the suggestion and we apologize to have accidentally missed this remark in the previous submitted version of the revised manuscript.

Best regards,

Michel SEVE

Round 3

Reviewer 2 Report

The paper is acceptable to Be published at the current form.